



# Determination of Aethalometer multiple-scattering enhancement parameters and impact on source apportionment during the winter 2017-2018 EMEP/ACTRIS/COLOSSAL campaign in Milan.

Vera Bernardoni[1,*], Luca Ferrero[2], Ezio Bolzacchini[2], Alice Corina Forello[1], Asta Gregorič[3,4], Dario Massabò[5], Griša Močnik[4,6], Paolo Prati[5], Martin Rigler[3], Luca Santagostini[2], Francesca Soldan[1,#], Sara Valentini[1], Gianluigi Valli[1], Roberta Vecchi[1]

[1]Dipartimento di Fisica "A. Pontremoli", Università degli Studi di Milano & INFN-Milan, 20133 Milano, Italy
[2]GEMMA and POLARIS Centre, Università degli Studi di Milano-Bicocca, 20126 Milano, Italy
[3]Aerosol d.o.o., Kamniška 39A, SI-1000 Ljubljana, Slovenia
[4]Center for Atmospheric Research, University of Nova Gorica, Vipavska 11c, SI-5270 Ajdovščina, Slovenia
[5]Dip. di Fisica Università di Genova & INFN Sezione di Genova, Via Dodecaneso 33, 16146 Genova, Italy
[6] Department of Condensed Matter Physics, Jozef Stefan Institute, Jamova 39, SI-1000 Ljubljana, Slovenia
[#]now at: Ricerca sul Sistema Energetico - RSE S.p.A., 20134 Milan, Italy

*Correspondence to:* Vera Bernardoni (vera.bernardoni@unimi.it)

**Abstract.** In the frame of the EMEP/ACTRIS/COLOSSAL campaign in Milan during winter 2018, equivalent black carbon measurements using the Aethalometer 31 (AE31), the Aethalometer 33 (AE33), and the Multi-Angle Absorption Photometer (MAAP) were carried out together with levoglucosan analyses on 12-h resolved $PM_{2.5}$ samples collected in parallel.

From AE31 and AE33 data, the loading-corrected aerosol attenuation coefficients ($b_{ATN}$) were calculated at 7 wavelengths ($\lambda$s, where $\lambda$ = 370, 470, 520, 590, 660, 880, 950 nm). Aerosol absorption coefficient at 637 nm ($b_{abs\_MAAP}$) was determined by MAAP measurements. Furthermore, $b_{abs}$ was also measured at 4 wavelengths (405, 532, 635, 780 nm) on the 12-h resolved $PM_{2.5}$ samples by a polar photometer (PP_UniMI).

After comparing PP_UniMI and MAAP results, we exploited PP_UniMI data to evaluate the filter multiple-scattering 25 enhancement parameter at different wavelengths for AE31 and AE33. We obtained instrument- and wavelength-dependent multiple-scattering parameters by linear regression of the Aethalometer $b_{ATN}$ against the $b_{abs}$ measured by PP_UniMI. We found significant dependence of the multiple-scattering enhancement parameter on filter material, hence on the instrument, with the difference up to 30% between the AE31 and the AE33 tapes. The wavelength dependence and day/night variations were small – the difference between the smallest and largest value was up to 6%.

Data from the different instruments were used as input to the so-called "Aethalometer model" for optical source apportionment and instrument-dependence of the results was investigated. Inconsistencies among the source apportionment were found fixing the AE31 and AE33 multiple-scattering enhancement parameters to their usual values. Opposite, optimised multiple-scattering enhancement parameters led to 5% agreement among the approaches.





Also, the component-apportionment "MWAA model" was applied to the dataset. It resulted less sensitive to the instrument

and the number of wavelengths, whereas significant differences in the determination of the absorption Ångström exponent

for brown carbon were found (up to 22%).

## 1. Introduction

Light absorbing aerosols are of great interest for their effects: they provide a positive radiative forcing at global scale (IPCC, 2013) and can affect visibility at local scale (see e.g. Valentini et al. (2018) for estimates in Milan).

Black carbon (BC) and brown carbon (BrC) are major light absorbing aerosol species. They differ both in the extent of light absorption per mass and its wavelength-dependence (Bond et al., 2013; Laskin et al., 2013). Furthermore, BC is a primary component and it is emitted in every incomplete combustion process. An important primary source of BrC is wood burning (e.g., Lack et al., 2013; Lu et al., 2015; Saleh et al., 2014; Washenfelder et al, 2015); recently, also other possible sources of BrC have been reported, e.g., BrC formation by secondary processes (Liu et al., 2015; Kumar et al., 2018). Mineral dust is

another possible light absorber. At mid latitudes, its contribution is generally episodic and related to desert dust transport episodes (e.g. Fialho et al., 2005).

Thus, aerosol absorption properties at different wavelengths are of interest not only to better characterise the interaction with solar radiation, but also as inputs to models for optical source apportionment using the Aethalometer model (Sandradewi et al, 2008) and for the identification of BC and BrC contribution to the absorption coefficient (component apportionment)

using e.g. the Multi-Wavelength Absorption Analyzer model (MWAA model, Massabò et al., 2015). Nevertheless, it must be recalled that particle absorption properties depend on particle size, composition, and mixing state. It is noteworthy that neither reference instruments (Bond et al., 2013; Moosmüller et al., 2009; Petzold et al., 2013) nor reference materials (Baumgardner et al., 2012) exist for the measurement of the aerosol absorption coefficient ($b_{abs}$). Thus, $b_{abs}$ measurement and apportionment are still burning open issues in aerosol science.

Among the approaches for $b_{abs}$ determination, filter-based measurements are widely used: indeed, filter-based automatic instruments (able to operate for months with no need of maintenance) provide $b_{abs}$ information with high temporal resolution with the advantage to obtain long-term data series of $b_{abs}$. Besides on-line devices, two off-line multi-wavelength instruments based on polar photometry were also developed in the last decade: the polar photometer PP_UniMI (Bernardoni et al., 2017a; Vecchi et al, 2014) and the Multi-Wavelength Absorption Analyzer MWAA (Massabò et al., 2013; Massabò et al.,

2015). All filter-based measurements are affected by multiple-scattering effects as the aerosol is collected on fibre filters, and by loading effects – i.e. non-linearities in light attenuation during filter loading (Liousse et al., 1993; Petzold et al., 1997; Bond et al., 1999; Moosmüller et al., 2009). Different approaches are used for the correction of loading and multiple-scattering effects in filter-based instruments (e.g. Drinovec et al. 2015; Petzold and Schönlinner, 2004; Virkkula et al, 2007; Virkkula, 2010; Weingartner et al., 2003), and the details for those considered in this work will be explained in section 2.2.

Notwithstanding such corrections, inter-comparability of different instruments for the determination of the aerosol



absorption properties is still an open methodological issue especially for ambient aerosol measurements. Among filter-based instruments, the Multi-Angle Absorption Photometer (MAAP) is generally considered as a reference (Ammerlaan et al., 2017; Müller et al., 2011) and off-line measurements carried out with analogous principle will be used in this work to provide contribution to the debate on the treatment of multiple-scattering effects for Aethalometers (Backman et al., 2017;

Collaud-Coen et al., 2010; Di Biagio et al., 2017; Kim et al., 2019; Laing et al., 2020; Müller et al., 2011; Saturno et al., 2017; Schmid et al., 2006; Segura et al., 2014; Valentini et al., 2020; Weingartner et al., 2003).

As previously mentioned, despite the problems concerning $b_{abs}$ measurements harmonisation, these data are used as input for optical source apportionment and component apportionment models. The most widespread among these models is the Aethalometer model (Sandradewi et al., 2008), which aims to apportion fossil fuel combustion (FF) and wood burning (WB)

contributions to $b_{abs}$. For both sources, representative absorption Ångström exponent ($\alpha_{FF}$ and $\alpha_{WB}$, respectively) are free parameters of the model and have to be chosen a priori. Plenty of literature was spent on difficulties related to the choice of these parameters (e.g. Harrison et al., 2013, Fuller et al., 2014; Helin et al., 2018, Martinsson et al., 2017, Zotter et al., 2017). On the contrary, much less attention was dedicated to the role of the instrument providing the input data on the output of the Aethalometer model. Similarly, no investigation on the role of the instrument providing input data to the MWAA model for

component apportionment is present in the literature.

This work tries to expand these fields and will show the results of the winter EMEP/ACTRIS/COLOSSAL campaign carried out in Milan in January and February 2018. Different filter-based on-line instruments were deployed (MAAP and Aethalometers mod. AE31 and mod. AE33), and sampling was carried out in parallel with 12-h resolution on quartz-fibre filters for the analysis by PP_UniMI. The work will show results about:

-   The assessment of multiple-scattering enhancement parameters at different wavelengths for AE31 and AE33 by comparison with off-line measurements by PP_UniMI, including possible wavelength-dependence and daytime vs. night-time differences.

-   The role of input data provided by different instruments in the output of the Aethalometer model and MWAA model.

**2. Methods**

**2.1 Sampling campaign**

The sampling campaign was carried out at an urban background station in Milan, on the roof of the U9 building of the University of Milan-Bicocca (45°30'38"N, 9°12'42"E, 10 m a.g.l.) in the frame of the EMEP/ACTRIS/COLOSSAL winter campaign. All the instruments/samplers were equipped with $PM_{2.5}$ size-selective inlets. Aethalometers mod. AE31 and mod.

AE33 (in the following named AE31 and AE33, respectively, Magee Scientific, Aerosol) sampled continuously from 16 January to 20 February 2018 with 5-minute and 1-minute temporal resolution, respectively. In addition, from 17 January to 16 February, a Multi-Angle Absorption Photometer (MAAP, Thermo-Fischer) was operated in parallel with 5-minute





temporal resolution. Moreover, fifty-seven 12-h resolved PM2.5 samples (h. 6-18, 18-6, LST local standard time) were collected using a sequential low-volume sampler (TCR-TECORA, Italy) at 1m³/h on pre-fired (700°C, 1h) 47-mm quartz

fibre filters (QAO-UP, Pall) for absorption coefficient off-line analyses.

### 2.2 Optical measurements

### 2.2.1. Aethalometers AE31 and AE33

The Aethalometers AE31 and AE33 perform on-line light-transmission measurements through a filter tape at 7 wavelengths (370, 470, 520, 590, 660, 880 and 950 nm). The output of both instruments at each wavelength ($\lambda$) is expressed as the

concentration of equivalent black carbon (eBC($\lambda$)) (Hansen et al., 1982; Petzold et al., 2013), as it is considered as the only absorber. Being based on light transmission measurements only, the multiple-scattering effect (optical path enhancement induced by both the filter and the sample, making complicated accounting for both) and filter loading effects (non-linear optical path reduction induced by absorbing particles accumulating on the filter) (Weingartner et al., 2003; Arnott et al., 2005; Collaud-Coen et al., 2010) have to be accounted for to retrieve information on aerosol light absorption.

For both AE31 and AE33, linear relationship as in Eq. (1) is assumed between the loading-corrected attenuation coefficient $b_{ATN}$ and the absorption coefficient $b_{abs}$ at a considered wavelength is assumed in the form:

$$b_{ATN} = C \cdot b_{abs} \qquad (1)$$

where C is named multiple-scattering enhancement parameter (see sections 2.2.1.1 and 2.2.1.2). The following paragraphs provide details of the operation principles of both AE31 and AE33.

*2.2.1.1 Aethalometer AE31.*

The Aethalometer AE31 collects ambient aerosol on a spot on a quartz filter tape (Pall Q250 quartz) and measures the

attenuation (ATN) at all available wavelengths:

$$ATN(\lambda) = -100 \cdot \ln(I(\lambda)/I_0(\lambda)) \qquad (2)$$

where in Eq. (2) $I_0$ is the intensity of light transmitted through the blank filter spot and I is the intensity measured at a

specific moment through the sampled spot.

To avoid the measurement of heavily loaded spot, the tape moves automatically to a fresh spot when ATN(370nm)=120.

For AE31, the loading effect can be compensated by different off-line algorithms, as proposed in the literature (see e.g. Arnott et al., 2005; Collaud Coen et al., 2010; Schmid et al., 2006; Virkkula et al., 2007; Weingartner et al., 2003). In this work, the loading effect was corrected by applying the Weingartner et al. (2003) procedure. Therefore, using the



measurements of the eBC provided by the AE31 at different wavelengths ($eBC_{AE31}(\lambda)$) and considering the default $\lambda$-dependent mass attenuation cross sections in use for the AE31 ($\sigma_{AE31}(\lambda)$), the loading-corrected attenuation coefficient ($b_{ATN\_AE31}(\lambda)$) was obtained as:

$$b_{ATN\_AE31}(\lambda) = R(ATN_{AE31}) \cdot eBC_{AE31}(\lambda) \cdot \sigma_{AE31}(\lambda) \qquad (3)$$


where the shadowing term $R(ATN_{AE31})$ in Eq. (3) was dynamically determined following the Sandradewi et al. (2008b) algorithm. The present approach was recognised to be one of the best ones as corrected data are in good agreement with measurements from the MAAP and correction does not affect data in terms of the absorption Ångström exponent (Collaud Coen et al., 2010). Implementing corrections not affecting the absorption Ångström exponent is of great importance e.g. in

heating rate studies; the approach mentioned above has been already applied successfully at the investigated site on data series starting from 2015 (Ferrero et al., 2018) and it will be performed in the companion paper by Ferrero et al. (submitting) on the same dataset.

As for the multiple-scattering enhancement parameter in Eq. (1), for AE31 $C_{AE31\_0} = 2.14$ was originally proposed by Weingartner et al., (2003). This value was already evidenced to be underestimated by comparison of $b_{ATN,AE31}$ with different

reference instruments (e.g. MAAP, photoacoustic spectrometers, extinction-minus-scattering technique): depending on the sampling site and methodology, values in the range 3-8 were reported (e.g. Backman et al., 2017; Collaud-Coen, 2010; Di Biagio et al., 2017; Kim et al., 2019; Müller et al., 2011; Saturno et al., 2017; Segura et al., 2014). Based on the previous literature, possible wavelength-dependence of the multiple-scattering enhancement parameters is another open issue. Currently, guidelines from the Global Atmosphere Watch Programme suggest the use of $C_{AE31}=3.5\cdot(1 \pm 0.25)$ (GAW, 2016).

For these reasons, one the objective of this work is its experimental assessment exploiting PP_UniMI measurements as explained in section 2.5. Considering that $eBC_{AE31}(\lambda)$ concentration is reported by the instrument at standard volumetric flow (20°C and 1013hPa). To allow comparison with PP_UniMI data (reported at ambient conditions and 12-h resolution), $eBC_{AE31}(\lambda)$ was firstly recalculated to the ambient flow conditions and then used to retrieve $b_{ATN\_AE31}(\lambda)$.

*2.2.1.2 Aethalometer AE33*

AE33 is the latest version of the Aethalometer. It collects ambient aerosol in parallel on two filter tape spots of the same area at different flowrates. In this work, the TFE-coated glass fibre filter tape T60A20 was used (Drinovec et al., 2015). Similarly to AE31, the tape is automatically moved to the fresh area of the tape to avoid heavily loaded spots. Highly time-resolved information on the light transmitted through the two spots at 7 different wavelengths is used to determine the loading-

corrected attenuation coefficient ($b_{ATN\_AE33}(\lambda)$) in real-time using the "dual spot" algorithm described in Drinovec et al. (2015).

Also for AE33, the output of the instrument is equivalent black carbon concentration at different wavelengths ($eBC_{AE33}(\lambda)$), but in this case two steps are needed to reconstruct the measured $b_{ATN\_AE33}(\lambda)$. Indeed:





- the instrument implements wavelength-dependent mass absorption cross sections (MAC($\lambda$)) which relate the eBC$_{AE33}$($\lambda$) to the aerosol absorption coefficient b$_{abs\_AE33}$($\lambda$) as in Eq. (4):

$$b_{abs\_AE33}(\lambda) = eBC_{AE33}(\lambda) \cdot MAC(\lambda) \tag{4}$$

- b$_{abs\_AE33}$($\lambda$) is related to b$_{ATN\_AE33}$($\lambda$) as in Eq. (1), where C$_{AE33\_0}$=1.57 was suggested by manufacturer for the filter tape in use for harmonisation to AE31 data.

As eBC$_{AE33}$($\lambda$) data are reported by the instrument at standard volumetric flow (21.1 °C and 1013.25 hPa), b$_{ATN\_AE33}$($\lambda$) were referred to ambient pressure and temperature (12-h average) to allow comparison with PP_UniMI data.

As done for AE31, experimental investigation on the suitability of C$_{AE33\_0}$ was performed as explained in section 2.5. Indeed, literature works point to C$_{AE33\_0}$=1.57 as underestimated. As examples, Valentini et al. (2020) identified C$_{AE33}$=2.66 as suitable in Rome by comparison of b$_{ATN\_AE33}$ vs. b$_{abs,MAAP}$ and Laing et al., 2020 report C$_{AE33}$=4.37 by comparison with suitably corrected tri-color absorption photometer (TAP) b$_{abs,TAP}$ measurements.

### 2.2.2 MAAP

The MAAP (637 nm, Müller et al., 2011) collects aerosol on a spot on a filter-tape and, as for the Aethalometers, the filter tape is suitably moved to avoid heavy loading when transmittance reaches a value that can be set by the user: in this work, default value (20%) was used. MAAP measures the light transmitted and scattered at fixed angles. Optimised analytical functions are used to retrieve the total light in the front and back hemispheres by solid-angle integration (Petzold and Schönlinner, 2004). The MAAP algorithm implements a suitable radiative transfer model accounting for particle-filter matrix interactions (Hänel, 1987; Hänel, 1994). Results obtained using this method directly correct for multiple-scattering effects and are no issue related to filter loading was observed (Petzold et al., 2005).

As reported in Petzold and Schönlinner (2004), the input to this model are:

- the ratios between the loaded and the blank spots analytical function integrals determined for the front and backward hemispheres, separately;
- backward-to-total light integral ratio for the blank filter matrix B$_M$ = 0.7
- asymmetry parameter g = 0.75.

The raw outputs of the model are the optical depth ($\tau$) and the single scattering albedo ($\omega$) of the filter layer containing the particles. The aerosol absorption coefficient (b$_{abs}$, expressed in Mm$^{-1}$) in atmosphere during the sampling is determined considering the deposit area (A in cm$^2$) and the sampled volume (V in m$^3$) as in Eq.(5):

$$b_{abs} = 100 \cdot (1-\omega)\tau\frac{A}{V} \tag{5}$$





Overall, a 12% uncertainty was reported (Petzold and Schönlinner, 2004). Assuming a constant mass absorption cross section (6.6 $m^2/g$), the output of the MAAP is the equivalent black carbon concentration in air ($eBC_{MAAP}$), expressed in $\mu g/m^3$. Further details on the instrument are reported in Müller et al. (2011).

### 2.2.3 PP_UniMI analyses

The aerosol absorption coefficient at 4 wavelengths (405 nm, 532 nm, 635 nm, 780 nm) was determined on the collected $PM_{2.5}$ samples using the polar photometer PP_UniMI at the University of Milan (Vecchi et al., 2014, Bernardoni et al., 2017a). In PP_UniMI, the chosen laser beam hits the filter (either blank or loaded) perpendicularly. The filter transmits and scatters light in the front and back hemispheres. A photodiode mounted on a rotating arm scans the scattering plane (0-173° with about 0.4° resolution) allowing the determination of the total amount of light diffused in the two hemispheres by solid

angle integration.

In usual PP_UniMI operation - hereinafter named "PP approach" (PP) - the same radiative transfer model as the one used in the MAAP is applied, but the following differences in input data evaluation have to be highlighted:

- front and backward hemisphere integrals are determined by solid angle integration of the high-angular resolution phase function measurements and not by analytical function integrals;

- no assumption on $B_M$ is done, as it is directly obtained by the measurements of the blank filter.

As well as for the MAAP, the outputs of the models are $\omega$ and $\tau$. The minimum detection limits on the absorbance (ABS=(1-$\omega$)·$\tau$) of the particle-containing layer of the samples are in the range 0.03-0.07 depending on the wavelength. It is also noteworthy that samples with ABS>0.9 were excluded by the database to avoid possible non-linearities due to sample overloading. Uncertainties were estimated in ±0.01 for ABS<0.1 and 10% for ABS ≥0.1 (Bernardoni et al., 2017a)

It is noteworthy that exploiting information at suitable angles, the same approximations used in the MAAP calculation can be implemented, i.e. total amount of light in the two hemispheres by analytical functions can be obtained, and $B_M$=0.7 can be imposed, for the sake of comparison. This approach will be in the following referred to as "PP_UniMI as MAAP" (PaM) approach.

In both approaches (PP and PaM), the aerosol absorption coefficient at all PP_UniMI measurement wavelengths ($b_{abs,PP}(\lambda)$

and $b_{abs,PaM}(\lambda)$ for PP and PaM, respectively) can be obtained from $\omega$ and $\tau$, considering the deposit area A=11.9 $cm^2$ and the total sampled volume using Eq. (5). The comparison between the two approaches will be carried out through Deming linear regressions, as explained in section 2.8.

### 2.3 Levoglucosan measurements

After being analysed by PP_UniMI, one punch (1.5 $cm^2$) of each 12-h sample was devoted to the measurement of

levoglucosan concentration. Each punch was extracted by sonication (1-h) using 5 mL ultrapure (Milli-Q) water. The analysis was carried out by High-Performance Anion Liquid Chromatography coupled with Pulsed Amperometric Detection (HPAEC-PAD) at the University of Genoa following the procedure described in Piazzalunga et al. (2010). Minimum





detection limit for levoglucosan is about 2 ng/ml (i.e. 6.6 ng/m$^3$ considering the filter area and sampling volume) and uncertainties are ~11%.

**2.4 Experimental absorption Ångström exponent**

The experimental absorption Ångström exponent ($\alpha_{exp}$) was determined for each 12-h time slot from all instruments fitting the parameters $K_{exp}$ and $\alpha_{exp}$ in Eq. (6):

$$b_{abs}(\lambda) = K_{exp}\lambda^{-\alpha_{exp}} \tag{6}$$

It is noteworthy that light absorbing components (e.g. BC vs. BrC) have different $\lambda$-dependences and they both contribute to $\alpha_{exp}$. Thus, it is not expected that Eq. (6) represents exactly the wavelength-dependence of the measurements (i.e. $\alpha_{exp}$ is expected - and renown - to be dependent on the range of wavelengths considered in the calculation). Anyway, it is a good approximation and it can be exploited to gain information at wavelengths different from the measured ones (see e.g. application in section 2.5).

**2.5 Optimisation of multiple-scattering enhancement parameters**

Optimised multiple-scattering enhancement parameters at 4 different wavelengths for AE31 and AE33 ($C_{AE31}(\lambda)$, $C_{AE33}(\lambda)$, respectively) were retrieved by comparing loading-corrected attenuation coefficients $b_{ATN\_AE33}(\lambda)$ with the absorption coefficient measured by PP_UniMI, with both PP and PaM approaches (section 2.2), through a Deming linear regression analysis explained in section 2.8. When the intercept of the regression was comparable to zero, the slope of the regression
line directly represented the best estimate for the corresponding multiple-scattering enhancement parameter.

To allow such comparison, PP_UniMI data were interpolated/extrapolated to Aethalometer wavelengths exploiting $\alpha_{exp}$ calculated as explained in section 2.4 through the following relationships:

$$b_{abs}(470nm)=b_{abs}(405nm)(470/405)^{-\alpha_{exp}}$$

$$b_{abs}(520nm)=b_{abs}(532nm)(520/532)^{-\alpha_{exp}}$$

$$b_{abs}(660nm)=b_{abs}(635nm)(660/635)^{-\alpha_{exp}}$$

$$b_{abs,}(880nm)=b_{abs}(780nm)(880/780)^{-\alpha_{exp}}$$

It was already demonstrated for Aethalometer data that exploiting information at 370 nm or 470 nm for the evaluation of the absorption Ångström exponent has important impact on the result whereas information at longer wavelengths plays a minor role (Zotter et al., 2017). For these reasons, no extrapolation of PP_UniMI data at wavelengths shorter than 405 nm was
performed; opposite, extrapolation was attempted at least at the nearer longer Aethalometer wavelength (i.e. 880 nm), as on that side the curve is less steep and possible biases are expected to be smaller.




To ensure consistent comparison of the results at different wavelengths, only samples for which PP_UniMI information was available at all wavelengths were considered (i.e. samples in which measurements at all wavelengths were higher than LOD and with ABS<90).

**2.6 Aethalometer model**

The Aethalometer model was introduced by Sandradewi et al. (2008). Generally, the model is used to apportion the contribution of fossil fuel combustion (FF) and wood burning (WB) to both the aerosol absorption coefficient ($b_{abs}$) and carbonaceous fractions. In this work, we will focus on the $b_{abs}$ source apportionment only. Please note that in this paragraph we will use $b_{abs}$ with no explicit reference to the instrument used for its determination as it does not affect the explanation of 265 the Aethalometer model itself.

The Aethalometer model exploits 2-$\lambda$ $b_{abs}$ measurements as input data and it is based on the following assumptions:

- at both wavelengths, FF and WB are the only sources contributing to the measured $b_{abs}$, as expressed in Eq. (7):

$$b_{abs}(\lambda) = b_{abs,FF}(\lambda) + b_{abs,WB}(\lambda) \qquad (7)$$

- for fossil fuel combustion, it holds Eq. (8):
$$\frac{b_{abs,FF}(\lambda_1)}{b_{abs,FF}(\lambda_2)} = \left(\frac{\lambda_1}{\lambda_2}\right)^{-\alpha_{FF}} \qquad (8)$$

where $\lambda_1$ indicates a short wavelength, $\lambda_2$ a long wavelength, and $\alpha_{FF}$ is a parameter assumed a-priori, representing the absorption Ångström exponent for the fossil fuel combustion source.

- for wood burning it similarly holds Eq. (9):
$$\frac{b_{abs,WB}(\lambda_1)}{b_{abs,WB}(\lambda_2)} = \left(\frac{\lambda_1}{\lambda_2}\right)^{-\alpha_{WB}} \qquad (9)$$

where $\alpha_{WB}$ is another parameter assumed a-priori representing the absorption Ångström exponent for wood
burning.

The identification of suitable $\alpha_{FF}$ and $\alpha_{WB}$ for the considered campaign/sampling site is recognised as the critical step in the modelling procedure and different approaches were proposed (e.g. Harrison et al., 2013, Fuller et al., 2014; Helin et al., 2018, Martinsson et al., 2017, Zotter et al., 2017, Forello et al., 2019, Forello et al., 2020); opposite, less attention was posed to the role of using data from different instruments as input to the model.





However, once $\lambda_1$, $\lambda_2$, $\alpha_{FF}$, $\alpha_{WB}$ are chosen, the $b_{abs}$ source apportionment at $\lambda_1$ and $\lambda_2$ is carried out combining Eq. (7) at $\lambda_1$ and $\lambda_2$, Eq. (8), and Eq. (9). Traditionally, the Aethalometer model is applied just considering the 470-950 nm wavelength pair. However, due to the purpose of the present work:

- for AE33 and AE31, the following wavelength pairs were considered: 470-880 nm, 370-950 nm, 370-880 nm, 470-950 nm;

- for PP_UniMI (with both PP and PaM approaches), only one test was performed using extreme values: 405 nm-780 nm.

It is noteworthy that AE31 and AE33 provide 7-$\lambda$ information, but the Aethalometer model represented by Eq. (7), Eq. (8), and Eq. (9) exploits information only at 2 chosen $\lambda$s (from now on named "2-$\lambda$ approach"). In this work, to exploit all the information provided by AE31 and AE33, we also propose an alternative approach, in the following named "multi-$\lambda$ fit".

The multi-$\lambda$ fit (regardless of the instrument) is based on Eq. (7) and keeps the $\lambda^{-\alpha_{FF}}$ and $\lambda^{-\alpha_{WB}}$ dependences reported in Eq.(8) and Eq.(9) for fossil fuel combustion and wood burning contributions, but these dependences are extended to all wavelengths, thus considering Eq. (10):

$$b_{abs}(\lambda) = A'\lambda^{-\alpha_{FF}} + B'\lambda^{-\alpha_{WB}} \tag{10}$$


Multi-wavelength fit of equation (10) is performed to retrieve the coefficients A' and B' for each sample, provided that values for $\alpha_{FF}$ and $\alpha_{WB}$ are defined a-priori. So, once A' and B' are determined for each sample and wavelength, $A'\lambda^{-\alpha_{FF}}$ represents the contribution of FF combustion to $b_{abs}(\lambda)$ and $B'\lambda^{-\alpha_{WB}}$ the WB one.

Of course, the available wavelengths depend on the considered instrument, and it is also possible test the method using

wavelength subsets. In this work, the whole available dataset (i.e. 4-$\lambda$: 405, 532, 635, 780 nm) was used as input for PP_UniMI (both in PP and PaM approaches), whereas for the Aethalometers both the use of all the 7 available wavelengths and of the 4 wavelengths for which multiple-scattering enhancement parameters were determined (i.e. 470, 520, 660, 880 nm) were tested, to analyse the role of extreme wavelengths. It is noteworthy that using our multi-$\lambda$ fit approach, it is possible to obtain the apportionment also at wavelengths different from the ones used as input (e.g. apportionment at

Aethalometer wavelengths using as input the data by PP_UniMI).

Focusing on Aethalometers, for all the 2-$\lambda$ and multi-$\lambda$ fit approaches tested, input $b_{abs}$ were obtained from Eq. (1) both using instrument-dependent $C_0$ and optimised multiple-scattering enhancement parameters presented in section 3.3 and obtained as reported in section 2.5.

In all tests, besides relative $b_{abs}(\lambda)$ source apportionment between FF and WB, correlation of $b_{abs,WB}$ with levoglucosan (in

terms of the Pearson correlation coefficient $r_{WB}$) was tested. Since no tracer in atmospheric aerosol for fossil fuel combustion was available, data on carbon monoxide (CO), nitrogen oxides ($NO_x$) and benzene concentrations from the Regional Environmental Protection agency database were tested as possible tracers for traffic emissions, which dominate fossil fuel





$b_{abs}$ contribution in Milan (Forello et al., 2019). Data were available at a traffic monitoring station at a distance about 2 km from our sampling site. Results of tests pointed to the benzene measurements at the traffic site as the best tracer for traffic, as

it showed the highest correlation with $b_{abs,FF}$ calculated for all instruments and calculation approaches (in terms of the Pearson correlation coefficient $r_{FF}$). Thus, correlation between benzene and $b_{abs,FF}$ will be shown. It is noteworthy that, thanks to the features of the model, $r_{FF}$ and $r_{WB}$ do not depend on the choice of the considered $\lambda$ for $b_{abs,FF}$ and $b_{abs,WB}$, respectively.

**2.7 MWAA model**

The MWAA model (Massabò et al., 2015; Bernardoni et al., 2017b) allows to assess the contributions of BC and BrC to the

total measured $b_{abs}(\lambda)$ (component apportionment), and to provide information on the absorption Ångström exponent for BrC ($\alpha_{BrC}$) exploiting Eq. (11):

$$b_{abs}(\lambda) = A\lambda^{-\alpha_{BC}} + B\lambda^{-\alpha_{BrC}} \qquad (11)$$

The coefficients A, B and $\alpha_{BrC}$ in equation (11) are obtained by multi-$\lambda$ fit of $b_{abs}(\lambda)$ for each sample, provided that a value for $\alpha_{BC}$ is assumed a-priori. In this case, $\alpha_{BC}=1$ was chosen as already performed in previous applications (Bernardoni et al., 2017b, Massabò et al., 2015).

Mathematically, at least 4-$\lambda$ measurements are needed to fit 3 parameters. Nevertheless, tests evidenced problems in numerical calculation when using only 4-$\lambda$ information (i.e. lack of convergence and/or fit parameter instability). Thus, in

this work we used at least 5-$\lambda$ information, consequently the MWAA model was run only using Aethalometer data as input. The fit of Eq. (11) was performed considering both the whole datasets (7-$\lambda$) and excluding extreme values (i.e. 5-$\lambda$: 470, 520, 590, 660, 880 nm) to gain insight into the role of the information at extreme wavelengths on the results. Fixed multiple-scattering enhancement parameters were considered, as the optimised ones were determined at 4-wavelengths only.

In section 3.3, the relative apportionment of the contributions from BC and BrC to $b_{abs}(\lambda)$ was shown. As the main

contributor to BrC is expected to be wood burning, the Pearson correlation coefficient ($r_{BrC}$) between the apportioned absorption coefficient for BrC ($b_{abs,BrC}$) and levoglucosan was also calculated. It is noteworthy that, as $\alpha_{BrC}$ is different for each sample, $r_{BrC}$ depends on the considered wavelength. As BrC is expected to provide higher relative contribution at decreasing wavelength, $r_{BrC}$ was presented at the shortest wavelength available in all test – i.e. $b_{abs,BrC}(470 \text{ nm})$ was used in $r_{BrC}$ evaluation.

**2.8 Deming regression**

In the results and discussion section (section 3), linear correlation between the data considered in the different comparisons were evaluated through the correlation coefficient *r*.



Linear regressions were performed using Deming regression (Deming, 1943; Ripley and Thompson, 1987). This approach is suitable when both data series are affected by not-negligible uncertainties (i.e. none of the series can be assumed as error-
free). The uncertainties associated to the data in the different cases will be described for each comparison.

The output of the Deming regression analysis will be represented in terms of slope, intercept, and their standard errors (SE). When the intercept of the Deming regression line was comparable to zero within 3-times the standard error (3·SE), it was forced through zero: in the text it will be reported "the intercept was comparable to zero" and only the slope of the intercept-forced regression will be presented. In the text and captions, "y vs. x" convention will be used (e.g. "PP vs. MAAP" means
that in the regression PP_Unimi data obtained with the PP approach were displayed on y axis and MAAP data on x axis).

### 3. Results and discussion

### 3.1 Comparison between MAAP and PP_UniMI results

The radiative transfer model used to account for multiple-scattering in the filter used for $b_{abs}$ determination by PP_UniMI (see section 2.2.3) was run using as input both PP and PaM approaches. It is noteworthy that, while PP approach fully
exploits highly angular-resolved measurements, PaM calculation introduces the same approximations as the ones used in the MAAP – i.e. reconstruction by analytical functions from measurements at 3 angles and the fixed value between backward and total diffused radiation for blank filter $B_M=0.7$ (section 2.2.2).

For each 12-h sample, $b_{abs,PP}(635\ nm)$ and $b_{abs,PaM}(635nm)$ were compared to the average 12-h $b_{abs,MAAP}$ (Figure 1). In both cases, high correlation is found (r > 0.991), and Deming regressions were performed with variance ratio = 1 (i.e. orthogonal
regression) as data had comparable uncertainties (see sections 2.2.2 and 2.2.3).

When exploiting all the available angular resolved information in the PP approach, the intercept was not comparable to zero ($-2.07 \pm 0.47$) and the slope was $0.928 \pm 0.021$. Nevertheless, comparing $b_{abs,PaM}(635\ nm)$ to the 12-h averaged $b_{abs,MAAP}$, the intercept was comparable to zero and the slope was $1.025 \pm 0.011$. The latter result confirms that PP_UniMI is equivalent to the MAAP when the same approximations were applied in calculation as performed in the PaM approach (section 2.2.3).
The previous comparisons also evidenced that the approximations implemented by the MAAP have a not-negligible impact on the measured $b_{abs,MAAP}$. The individual role of the phase function reconstruction and imposition of $B_M = 0.7$ is beyond the aim of the present work and it will be reported elsewhere (Valentini et al., in preparation), but first results indicate that the assumption on $B_M$ is the main responsible for the discrepancies. As for the presence of the intercept, this needs to be further investigated: scattering (Müller et al., 2011) or different penetration of the absorbers in the filter have been demonstrated to
produce spurious absorption signals (Arnott et al., 2005) at least for Aethalometers.

### 3.2 Comparison between PP and PaM approaches at all wavelengths

At wavelengths other than 635 nm, no comparison with MAAP is possible, thus only the comparison between the $b_{abs,PP}(\lambda)$ and $b_{abs,PaM}(\lambda)$ was performed. At all wavelengths, the results obtained were highly correlated (correlation coefficient r >





0.993), but significant deviation from 1:1 relation was found, with PP results generally lower than PaM ones. Focusing on

Deming regression line parameters (with variance ratio = 1), negative intercept was always found, whose absolute value

reduced with increasing wavelengths (see Table 1). In all cases, slope is not comparable to 1 within 3·SE.

### 3.3 Evaluation of multiple-scattering enhancement parameters for AE33 and AE31 during the campaign

PP_UniMI data were reported to Aethalometer wavelengths and used to gain information on multiple-scattering

enhancement parameters for AE33 and AE31 at different wavelengths ($C_{AE33}(\lambda)$, $C_{AE31}(\lambda)$, respectively) as explained in

section 2.5. In the following, results will be presented by comparing loading-corrected 12-h averaged $b_{ATN}(\lambda)$ from each

Aethalometer to both $b_{abs,PP}(\lambda)$ and $b_{abs,PaM}(\lambda)$. This was done because PP results are obtained with less assumptions than

those required by PaM approach. Nevertheless, PaM results were already demonstrated to be comparable to MAAP ones

(section 3.1), thus C-values obtained with this approach are more directly comparable to data commonly obtained by

research groups working with Aethalometers and MAAP in parallel for ambient measurements at urban or background

stations. The need to show both results highlights the importance of identifying a suitable reference material and reference

instrumentation.

Very high correlation (r>0.98) was found at all wavelengths between Aethalometers $b_{ATN}$ and both $b_{abs,PP}$ and $b_{abs,PaM}$.

Deming regression was performed considering the following uncertainties: a constant 1 $Mm^{-1}$ uncertainty was considered for

all instruments, summed to 10% uncertainty for PP_UniMI and increased to 15% for Aethalometers (as the effect of variable

aerosol scattering coefficient on the measurements is not considered).

In Fig. 2, scatterplots of the AE33 data against both PP (left panels) and PaM (right panels) approaches were shown at the

four wavelengths considered for comparison. In each scatterplot, lighter dots refer to daytime data, whereas the darker dots

refer to night-time data. Deming regression line on the whole dataset (day and night data) was also shown. Intercept of the

regression line was comparable to 0 at all wavelengths when calculated using the PaM approach data. In this case, the slope

of the regression line represented an average value for $C_{AE33\_PaM}$ and resulted in the range $2.78 \leq C_{AE33\_PaM}(\lambda) \leq 2.93$. These

values are about 10% higher than $C_{AE33}=2.66$ reported for Rome by Valentini et al. (2020) by comparison between AE33 and

MAAP (with no wavelength adjustment). Considering the PP calculation approach, the intercept was not comparable to zero

at 470 nm and 880 nm. Thus, we could provide $C_{AE33\_PP}(\lambda)$ from the regression slope only at 520 nm and 660 nm: we found

$C_{AE33,PP}(520 \text{ nm}) = 3.53 \pm 0.04$ and $C_{AE33\_PP}(660 \text{ nm}) = 3.37 \pm 0.05$. It is noteworthy that the approach presented in Eq. (1)

neglects a possible additive contribution from scattering (i.e. is best at low single scattering albedo). The presence of an

intercept not comparable to zero may indicate failure in such approximation. Furthermore, it has to be considered that few

$Mm^{-1}$ represent the limit of detection for PP_UniMI, thus it may have a role on the intercept.

Deming regression results were presented separately for daytime and night-time data in Table 2 for AE33. For these data, the

intercept of the regression line was comparable to zero. Exceptions were PP night-time results at 470 nm and 880 nm for

which the intercept exceeded 3·SE for less than 10% and they were forced the same. Daytime $C_{AE33}(\lambda)$ values were higher





than the corresponding night-time ones, even if they were comparable within SE for both PP and PaM calculation approaches. More in detail, multiple-scattering enhancement parameters calculated with PP approach were in the range $3.41 \leq C_{AE33,PP,day}(\lambda) \leq 3.57$ for daytime dataset and $3.31 \leq C_{AE33,PP,day}(\lambda) \leq 3.50$ for night-time dataset; calculations with the PaM approach gave $2.79 \leq C_{AE33,PaM,day}(\lambda) \leq 2.95$ for daytime dataset and $2.77 \leq C_{AE33,PaM,day}(\lambda) \leq 2.91$ for the night-time dataset. It is noteworthy that values at 470 nm and 520 nm are comparable within SE and the same occurs for the values at 660 nm and 880 nm for both PP and PaM approaches.

Figure 3 provides the same representation already explained in Fig. 2, considering in this case the AE31 dataset. All intercepts of the Deming regression carried out on the whole AE31 data were comparable to zero. In this case, it resulted $3.47 \leq C_{AE31\_PaM}(\lambda) \leq 3.58$ and these values were fully comparable to the suggested value of $3.5 \cdot (1 \pm 0.25)$ (GAW, 2016). Considering the PP approach, $4.22 \leq C_{AE31\_PP}(\lambda) \leq 4.33$ was found. It is noteworthy that for both $C_{AE31\_PP}(\lambda)$ and $C_{AE31\_PaM}(\lambda)$, the values at different wavelengths were comparable within SE, thus no statistically significant $\lambda$-dependence was observed.

Focusing on daytime and night-time datasets, separately, also for AE31 daytime $C_{AE31}(\lambda)$ values were higher than the corresponding night-time ones even if they were comparable within SE, considering both PP and PaM calculation approaches (see Table 3). More in detail, multiple-scattering enhancement parameters calculated with PP approach were in the range $4.34 \leq C_{AE33,PP,day}(\lambda) \leq 4.44$ for daytime dataset and $4.12 \leq C_{AE33,PP,night}(\lambda) \leq 4.25$ for night-time dataset; calculations with the PaM approach gave $3.55 \leq C_{AE33,PaM,day}(\lambda) \leq 3.65$ for daytime dataset and $3.39 \leq C_{AE33,PaM,night}(\lambda) \leq 3.53$ for the night-time dataset. For AE31, values at the different wavelengths were all comparable within SE for each approach, evidencing negligible $\lambda$-dependence.

It is noteworthy that all the $C_{AE31}(\lambda)$ values found comparing AE31 data with results by both PP and PaM approaches were higher than the corresponding values for AE33. This was expected, due to the different tape in use (recall $C_{AE31\_0}=2.14$ and $C_{AE33\_0}=1.57$ for the tapes in use).

Furthermore, multiple-scattering enhancement parameters calculated using $b_{abs,PP}(\lambda)$ as reference measurement for the absorption coefficient were always higher than those obtained using $b_{abs,PaM}(\lambda)$ as reference. This is due to the difference in the results by the two approaches evidenced in section 3.2, related to the approximations performed by the MAAP in the evaluation of the input to the radiative transfer model (see sections 2.2.2 and 2.2.3).

Last, it is noteworthy that both for AE33 data in Table 2 and AE31 data in Table 3, PaM values are 17-18% lower than the corresponding PP values. This seems higher than the slope reported in Table 1 (about 0.87-0.88), but a not-negligible negative intercept is also present, thus the global difference between the approaches is indeed higher than the value given by the slope.

### 3.4 Insights into $\alpha_{exp}$

For each 12-h time slot, Eq. (6) was exploited to calculate $\alpha_{exp}$ using as input $b_{abs}(\lambda)$ at all available wavelengths from AE31, AE33, and PP_UniMI with both PP and PaM approaches. In Fig. 4, frequency distribution of the calculated $\alpha_{exp}$ considering


wavelength-independent C values ($C_{AE31\_0}$=2.14 and $C_{AE33\_0}$=1.57 for AE31 and AE33, respectively), to obtain $b_{abs}(\lambda)$ from $b_{ATN}(\lambda)$ using Eq. (1).

Figure 4 showed that $\alpha_{exp}$ frequency distribution was narrower for Aethalometers datasets (1.1<$\alpha_{exp}$<1.8) than for PP_UniMI datasets in both PP and PaM approaches (0.9<$\alpha_{exp}$<2). Focusing on Aethalometers, AE31 distribution is more skewed towards lower values (with a sharp maximum bin in the 1.3-1.4 range) than AE33 distribution which is more symmetric.

These graphs immediately show that different λ-dependence is present in data from different instruments.

It is also of interest to gain insights into the effect of applying different multiple-scattering enhancement parameters to the data from AE31 and AE33 on the measured $\alpha_{exp}$. It should be recalled that in section 3.3 optimised multiple-scattering enhancement parameters were obtained at 470, 520, 660, 880 nm, only. So, $\alpha_{exp}$ from AE31 and AE33 data were re-calculated after evaluating $b_{abs}(\lambda)$ from Eq. (1) only at 470, 520, 660, 880 nm, with the following choices for the multiple-

scattering enhancement parameters:

1) at all wavelengths $C_{0\_AE31}$=2.14, $C_{0\_AE33}$=1.57 were considered;

2) day-time and night-time wavelength-dependent multiple-scattering enhancement parameters C, reported in Table 2 for AE33 and in Table 3 for AE31 were used. Both PP- and PaM-derived multiple-scattering enhancement parameters were considered. These values will be in the following named "optimised multiple-scattering

enhancement parameters".

Results of the $\alpha_{exp}$ frequency distributions obtained from these tests were shown in Fig. 5.

It is noteworthy that Fig. 5a and Fig. 4c as well as Fig. 5b and Fig. 4d differed only for the number of wavelengths used for $\alpha_{exp}$ calculation. The comparison confirmed the role of the chosen wavelengths on $\alpha_{exp}$ calculation, as already mentioned in section 2.4. More in detail, considering a narrower range of wavelength, $\alpha_{exp}$ distributions were narrower and peaked at lower

values.

The comparison of Fig. 5c and 5d to Fig. 4a as well as Fig. 5e and 5f to Fig. 4b, showed that the use of optimised multiple-scattering enhancement parameters was not enough to harmonise the results of $\alpha_{exp}$ from different instruments. There are different reasons for this. First of all, it is renown that experimental data are the sum of (at least) two contributions featuring different absorption Ångström exponents, thus the dependence is not expected to be exactly exponential; second, cross-

sensitivity to scattering is expected giving an additive term, which is neglected in the approach presented in Eq. (1) which approximates the relationship between absorption and extinction by the use of a single multiplicative factor. Last, we are considering average factors and applying them to all the dataset, whereas sample-by-sample differences e.g. in the scattering properties of the particles are expected. Finally, it should be recalled that PP_UniMI wavelengths were 405, 532, 635 and 780 nm, whereas the wavelengths considered for Aethalometers 4-λ calculations were 370, 520, 660, and 880 nm.

**3.5 Aethalometer model results**

As mentioned in section 2.6, multi-wavelength information on the aerosol absorption coefficient can be used as input to the Aethalometer model for source apportionment. Section 3.4 showed differences in the λ-dependences of data from different





instruments, as well as the impact of considering fixed or optimised multiple-scattering enhancement parameters. These observations point to the need of investigating the role of such differences on source apportionment results. So, in this
paragraph it will be investigated:

- the role of performing the Aethalometer model using data from different instruments
- the impact of applying wavelength-dependent multiple-scattering enhancement parameters on the Aethalometer model source apportionment results.

In this work, the Aethalometer model was run applying $\alpha_{FF}=1$ and $\alpha_{WB}=2$. These values were previously used in Bernardoni
et al. (2017b) for the Milan area during an application to a dataset with available wavelength information in the range 375-850 nm.

In the following, we will show results of the Aethalometer model run using as input data $b_{abs,PP}(\lambda)$, $b_{abs,PaM}(\lambda)$, and $b_{abs,AE31}(\lambda)$ and $b_{abs,AE33}(\lambda)$ obtained using both fixed multiple-scattering enhancement parameters and the optimised ones presented in section 3.3. Both the 2-$\lambda$ and the multi-$\lambda$ fit approaches (with all the possible combinations explained in section 2.6) were
tested. A summary of the average apportionment, correlation coefficients between the apportioned wood burning $b_{abs,WB}$ and levoglucosan measurements ($r_{WB}$), and correlation coefficients between the apportioned fossil fuel combustion $b_{abs,FF}$ and benzene measurements ($r_{FF}$) obtained with all the approaches was reported in Table 4.

From Table 4 and considering fixed multiple-scattering enhancement parameters for Aethalometers, it can be noted that:

1) Average apportionment percentage for AE31 and AE33 agreed within 7%, provided that the same short
wavelength was used as reference (either 370 nm or 470 nm), regardless of the data processing approach. Considering the same instrument, an average apportionment difference up to 12% was found at 470 nm for AE33 using 7-$\lambda$ approach compared to 2-$\lambda$ 470/950 nm. In any case, 7-$\lambda$ apportionment is never in the range of variability found considering 470 nm as lowest wavelengths, still evidencing the impact of near-UV measurements on the source apportionment results.

2) Average PP_UniMI apportionment was within 6% considering all approaches, and within 3% considering results from 4-$\lambda$ fit. Thus, it should be mentioned that – even if we evidenced significant differences in absolute values for PP and PaM measurements in section 3.2 – such differences do not impact significantly PP_UniMI relative source apportionment.

3) Correlation coefficients $r_{WB}$ between $b_{abs,WB}$ and levoglucosan showed high correlation ($r_{WB}\geq0.92$) for
AE33 and AE31 results, independently of the approach; opposite, lower correlation was found with all the PP_UniMI approaches ($r_{WB}\leq0.83$). Further investigation is needed to understand the reasons for this. This effect was possibly related to the wider $\alpha_{exp}$ frequency distribution found in section 3.4 for PP_UniMI data. Indeed, due to the fewer assumptions in $b_{abs}$ retrieval, PP_UniMI seems more sensitive than Aethalometers to sample-by-sample variability. Consequently, the approach of the Aethalometer model based on fixing
unique values of $\alpha_{FF}$ and $\alpha_{WB}$ for the whole dataset can make it less suitable to the application to such data. Nevertheless, this needs further investigation e.g. using multi-wavelength Nephelometers in parallel to

Aethalometers to perform more accurate corrections of Aethalometer data. It should also be evidenced the role of a single point affecting the correlation. It does not result as an outlier looking at wavelength $b_{abs}$ distribution, but its removal from the population increases $r_{WB}$ to 0.85-0.86, depending on the considered

approach.

4)   Correlation coefficients $r_{FF}$ between $b_{abs,FF}$ and benzene are in the range 0.87-0.92 (being slightly higher for Aethalometers), showing lower dependence on the instrument and/or approach than $r_{WB}$.

Table 4 also allowed to perform comparison between Aethalometer apportionment obtained using fixed or optimised multiple-scattering enhancement parameters. As an example, considering input data in the range 470-880 nm, AE31 and

AE33 $b_{abs,FF}$ relative contributions at 470 nm were in the range 59-65% considering fixed multiple-scattering enhancement parameters and 67-70% in the case of optimised ones; similarly, also considering other wavelengths for comparison, the ranges do not overlap. Thus, even if wavelength variabilities of multiple-scattering enhancement parameters were mostly within SE, they resulted in a significant impact on the average source apportionment results. Furthermore, PP_UniMI apportionments showed higher FF contributions than those obtained by AE31 and AE33 using fixed multiple-scattering

enhancement parameters (up to 7% when considering 470 nm as lowest wavelength for Aethalometers and up to 17% when comparing 7-$\lambda$ fit on AE33, again evidencing the important impact of the shortest wavelength on the source apportionment); opposite, relative apportionment agreed within 5% at most (and, more in detail, PP_UniMI source apportionments results were always within the variability of Aethalometers results by different approaches) when optimised multiple-scattering enhancement parameters were considered for Aethalometers.

This is an interesting result. Indeed, section 3.4 showed that the application of optimised multiple-scattering coefficient did not lead to fully harmonised $\alpha_{exp}$ frequency distributions. Nevertheless, here we showed that the use of optimised multiple-scattering parameters can lead to the harmonisation at least of the average relative source apportionment.

### 3.6 MWAA model results

As explained in section 2.7, the MWAA model for component apportionment was run using as input both 7-$\lambda$ and 5-$\lambda$ AE31 and AE33 data. In Table 5, relative contributions of BC and BrC to $b_{abs}(\lambda)$ obtained from the different tests was shown, together with $\alpha_{BrC}$ (average ± standard deviation) and $r_{BrC}$. Only Aethalometer wavelengths present also in Table 4 were reported.

Table 5 showed that the component apportionment performed by the MWAA model is less sensitive to extreme wavelengths

than the source apportionment performed by the Aethalometer model. Indeed, highest discrepancy of 5% in component apportionment and $r_{BrC} \geq 0.91$ were found at 470 nm in all cases. This was probably related to the ability of the model to self-evaluate the most suitable value for $\alpha_{BrC}$ as a function of input data. This was supported by the investigation of the role of different input data (in terms of instrument and wavelength range) on the computed $\alpha_{BrC}$. In Fig. 6, frequency distributions of $\alpha_{BrC}$ obtained in the different tests were shown: narrower distributions were obtained for AE33 than for AE31. This



observation held both for distributions obtained at 7-λ (Fig. 6a and 6b) and at 5-λ (Fig. 6c and 6d) and was confirmed considering that standard deviations of $\alpha_{BrC}$ values (Table 5) are 1.4 and 1.8 times higher for AE31 than for AE33. As for average $\alpha_{BrC}$ values, the role of the considered instrument or number of wavelengths is unclear. Indeed, average $\alpha_{BrC}$ obtained by AE33 data was 13% higher and 14% lower than those obtained by AE31 considering 7-λ and 5-λ, respectively. Furthermore, $\alpha_{BrC}$ computed at 7-λ was 18% lower and 7% higher than the one computed at 5-λ for AE31 and AE33,

respectively.

**Conclusions**

In this work, results from the EMEP/ACTRIS/COLOSSAL campaign carried out in Milan in winter 2018 were presented. The work explored some open issues in the measurements of the aerosol absorption coefficient by filter-based instrumentation and their impact on source (fossil fuel combustion/wood burning) and component (BC/BrC) apportionment.

Thanks to the comparison with off-line measurements carried out by the polar photometer PP_UniMI which performs high angular-resolved measurement of the sample phase function, we showed that the approximation introduced by the MAAP in the calculation can have a not-negligible impact on the results. Nevertheless, PP_UniMI and MAAP were demonstrated to provide comparable results when the same approximations were applied.

Furthermore, we exploited 4-wavelength $b_{abs}(\lambda)$ measurements carried out off-line by PP_UniMI to determine optimised

multiple-scattering enhancement parameters at different wavelengths for Aethalometers AE31 and AE33 - $C_{AE31}(\lambda)$ and $C_{AE33}(\lambda)$, respectively - by comparison with loading-corrected $b_{ATN,AE31}(\lambda)$ and $b_{ATN,AE33}(\lambda)$. $C_{AE31}(\lambda)$ and $C_{AE33}(\lambda)$ were calculated using PP_UniMI data obtained by considering both the whole high-angular resolved information – $b_{abs,PP}(\lambda)$, and using the approximations set in the MAAP – $b_{abs,PaM}(\lambda)$. Considering all AE31 samples compared to the PaM approach, $C_{AE31,PaM}(\lambda)$ results were in the range 3.47-3.58 and were comparable to the values prescribed by WMO/GAW (3.5 ± 25%).

As for AE33, $2.78 \leq C_{AE33,PaM}(\lambda) \leq 2.93$ depending on the wavelength was found from the PaM approach. Nevertheless, PP approach indicated that higher values (up to $C_{AE31,PP}(470nm)=4.33$ and $C_{AE33,PP}(520nm)=3.53$) can be more suitable, highlighting the role of MAAP approximations on the measured $b_{abs}$, but intercepts not comparable to zero were found in few cases, preventing the determination of an average value at 405 nm and 780 nm for AE33. This problem was overcome considering daytime and night-time data separately. In this case, daytime values of optimised multiple-scattering

enhancement parameters were slightly higher than the night ones, but within the standard error, for both AE31 and AE33 as well as using PP and PaM approach. Furthermore, also considering separately daytime and night-time data, values at different wavelengths were within SE for the same calculation approach. Separated daytime/night-time optimised multiple-scattering enhancement parameters were used for further investigation.

The analysis of the experimental absorption Ångström exponents ($\alpha_{exp}$) evidenced that significantly different values were

obtained depending both on the instrument and on the chosen wavelength-ranges from the same instruments. Wavelength-



dependent multiple-scattering enhancement parameters determined in this work were also applied to data from AE31 and AE33, but they were not enough to harmonise $\alpha_{exp}$ frequency distributions from different instruments.

This work investigated the role of such differences on the results of source apportionment by the Aethalometer model (by fixing a value of $\alpha_{FF}=1$ and $\alpha_{WB}=2$ already used in previous works in the area) and of the component appotionment by the

MWAA model (fixing $\alpha_{BC}=1$). The Aethalometer model was applied using as input $b_{abs}$ data determined by PP_UniMI, AE31 and AE33. As for AE31 and AE33, $b_{abs}(\lambda)$ obtained both using fixed and optimised multiple-scattering enhancement parameters were used as input. The role of different choices for the considered wavelength was also investigated, as well as different calculations approaches. Inconsistencies in relative source apportionment were found also considering a single instrument, evidencing not only the role of the chosen wavelength range (already found in the literature) but also that small

differences (within uncertainties) in the wavelength-dependencies of multiple-scattering enhancement parameters affect significantly the output of the Aethalometer model. Significant differences were found between the apportionment results from PP_UniMI data and those obtained by AE31 and AE33 with fixed values for the multiple-scattering enhancement parameters. It is noteworthy that the application of optimised multiple-scattering enhancement parameters did not harmonise $\alpha_{exp}$ frequency distributions among different instrument, but it led to consistent source apportionment results.

Focusing on the MWAA model, due to the features of the model our tests were limited to the assessment of the role of extreme wavelengths on the model results for AE31 and AE33. The average apportionment of the relative contributions of BC and BrC from AE31 and AE33 showed little influence on the considered wavelength range (5% maximum, to be compared to 11% limiting Aethalometer model analysis to the tests comparable to those performed by the MWAA model). Nevertheless, open issues remain concerning the estimates of $\alpha_{BrC}$, whose average value was in the range 2.99-3.66

depending on the instrument and the wavelength range considered as input.

**Acknowledgement**

The authors are grateful to Aerosol d.o.o for the availability of AE33 and to the Environmental Protection Agency of Lombardy Region (ARPA Lombardia) for the availability of benzene data.

This work was partially funded by the National Institute of Nuclear Physics (INFN), in the frame of the INFN-TRACCIA experiment.

This work was carried out with the GEMMA center support in the framework of Project MIUR "Dipartimenti di Eccellenza 2018-2022"

This work was carried out in the frame of the activities of the COST-COLOSSAL action CA16109 Chemical On-Line

cOmpoSition and Source Apportionment of fine aerosol.



**Author contribution**

**Conceptualisation**: V.B., L.F., E.B., G.V., and R.V. designed and organised the sampling campaign to meet the final goals. V.B. developed the data analysis strategy for optimized multiple-scattering enhancement parameter retrieval and developed the multi-λ fit approach for the Aethalometer model.

**Data curation**: V.B. and L.F. validated and assembled the final database

**Formal analysis:** V.B., L.F., A.C.F., G.M., L.S., S.V., G.V. collaborated to data analysis and reduction.

**Methodology:** V.B., L.F., G.V., R.V. realized the sampling campaign. A.G., M.R., G.M. gave support for Aethalometers set-up. F.S., S.V., and A.C.F. performed PP_UniMI measurements. D.M. and P.P. performed levoglucosan measurements.

**Software:** V.B. developed the software for multi-λ fit implementation of the Aethalometer model

**Supervision:** R.V. supervised all the scientific activity

**Writing – original draft:** V.B. wrote the original draft

**Writing – review & editing:** all co-authors commented and contributed to the final version of the paper

**Competing interest**

The authors declare that they have no conflict of interest. G.M. was employed at the manufacturer of the AE33 before the
start of this work. At the time of the research, A.G. and M.R. employed by the manufacturer of the Aethalometer instruments.

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



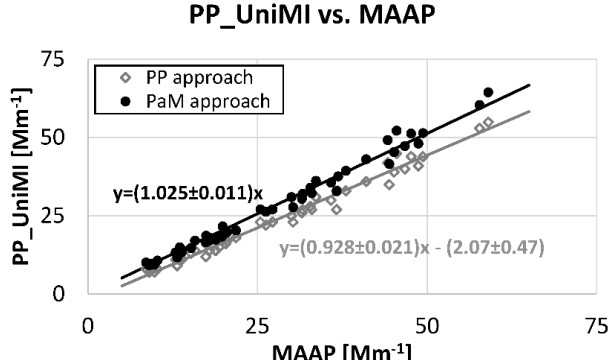

**Figure 1. Scatterplot of PP_UniMI data obtained using PP and PaM approach vs. MAAP.**

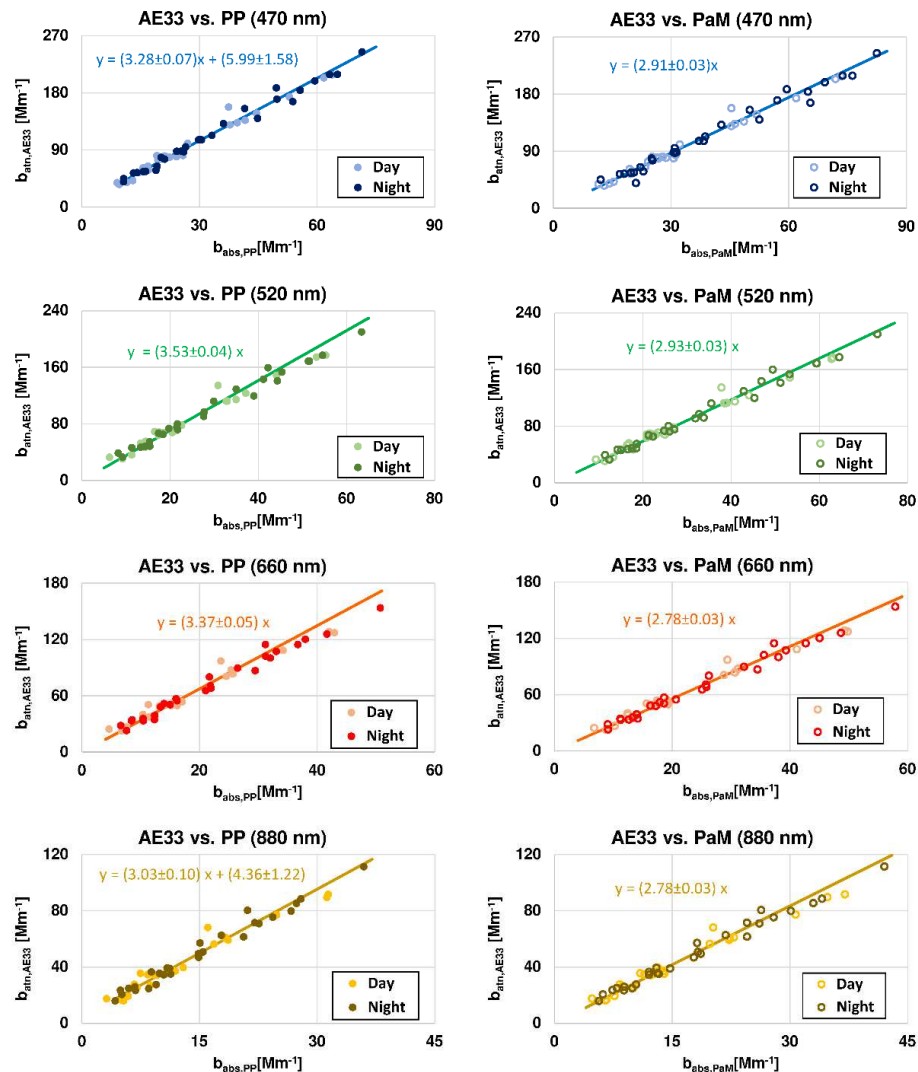

**Figure 2. Scatterplot of $b_{ATN\_AE33}(\lambda)$ vs. $b_{abs\_PP}(\lambda)$ (left charts) and $b_{abs\_PaM}(\lambda)$ (right charts) at 470 nm, 520 nm, 660 nm, and 880 nm (from top to bottom).**



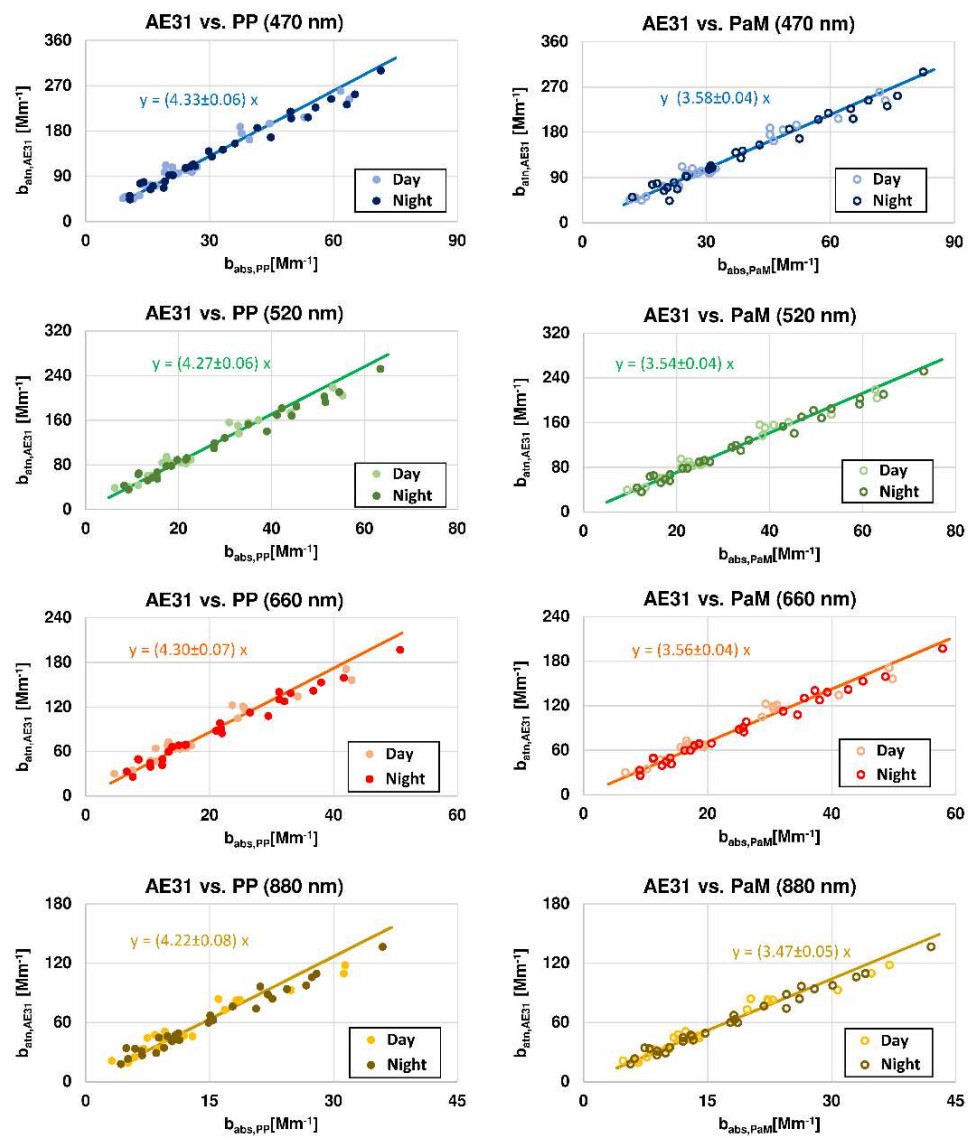


**Figure 3. Scatterplot of $b_{ATN\_AE31}(\lambda)$ vs. $b_{abs\_PP}(\lambda)$ (left charts) and vs. $b_{abs\_PaM}(\lambda)$ (right charts) at 470 nm, 520 nm, 660 nm, and 880 nm (from top to bottom).**





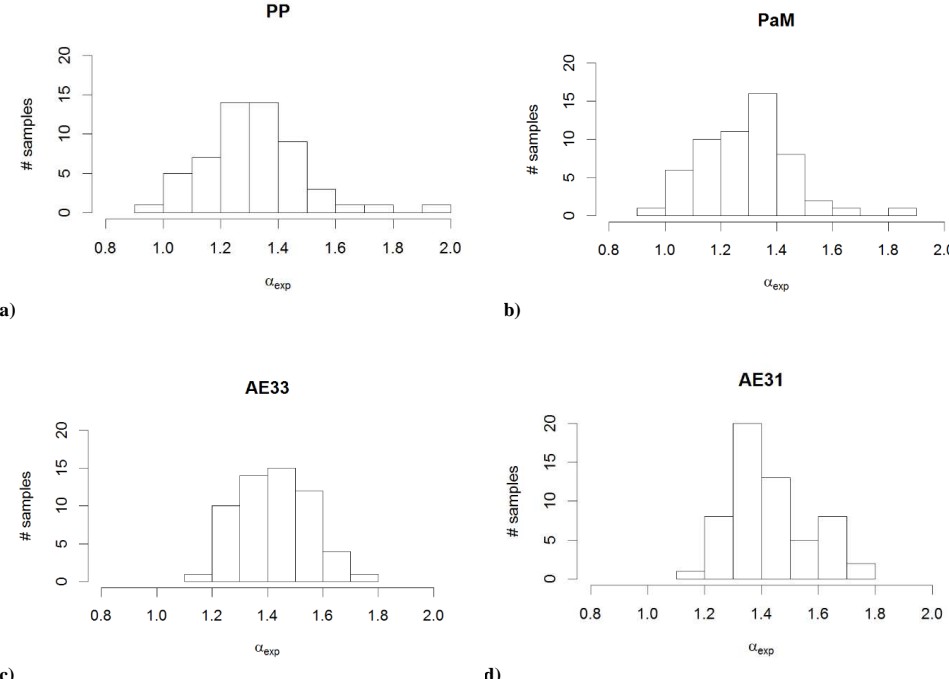

**Figure 4.** Frequency distribution in terms of number (#) of samples of the experimental absorption Ångström exponent ($\alpha_{exp}$) calculated using Eq. (6) for the different instruments. PP and PaM are calculated from 4-$\lambda$ $b_{abs}(\lambda)$ information in the range 405-780 nm, whereas AE33 and AE31 results are calculated from 7-$\lambda$ $b_{abs}(\lambda)$ obtained using $C_{AE31\_0}$ and $C_{AE33\_0}$ in the range 370-950 nm.





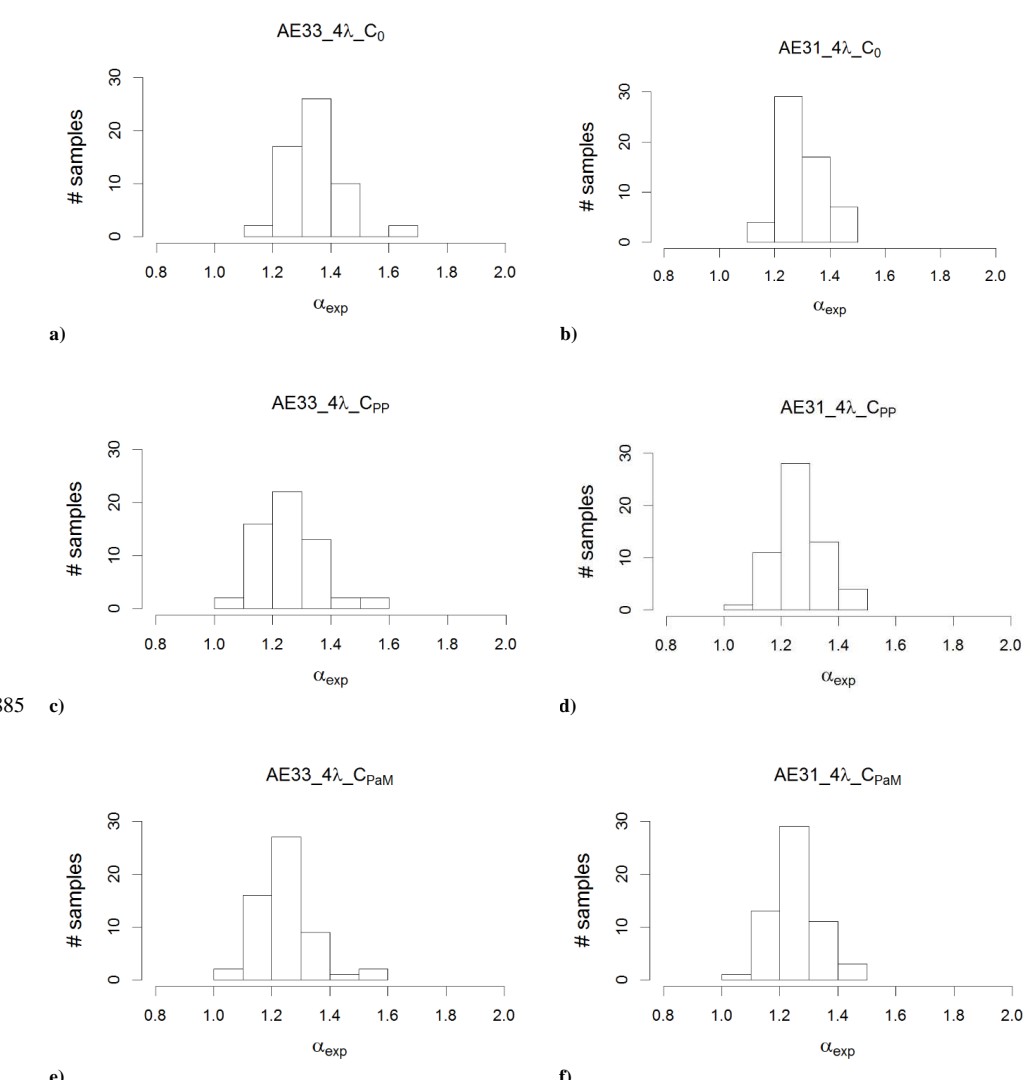


Figure 5. Frequency distribution of $\alpha_{exp}$ calculated from $b_{abs}(\lambda)$ at 470, 520, 660 and 880 nm for AE33 (left panels) and AE31 (right panels). The $b_{abs}(\lambda)$ to be fitted were obtained from Eq. (1) with the following choices for the multiple-scattering enhancement parameters: $C_{0\_AE33}$ and $C_{0\_AE31}$ in panels a) and b), data in Table 2 for panels c) and e), data in Table 3 for panels d) and f)






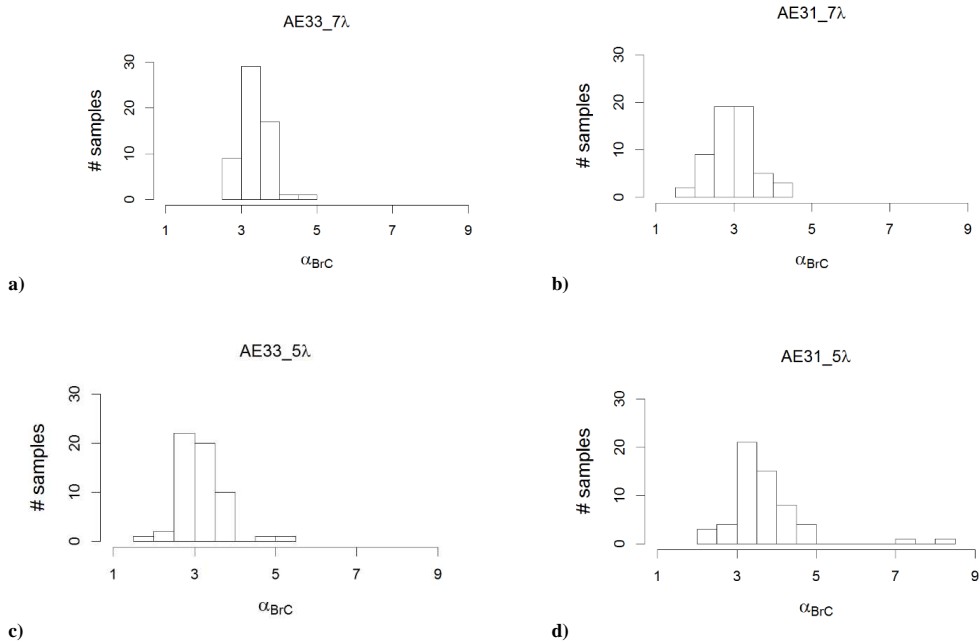

**Figure 6. Frequency distribution of α$_{BrC}$ computed by MWAA model for AE33 (left panel) and AE31 (right panel). Seven-wavelength b$_{abs}$ fit was performed in panels a) and b), and 5-wavelength fit was performed in panels b) and d).**






| Wavelength | Slope | SE slope | Intercept | SE Intercept |
|---|---|---|---|---|
| **405 nm** | 0.877 | 0.008 | -1.787 | 0.400 |
| **532 nm** | 0.878 | 0.006 | -1.284 | 0.190 |
| **635 nm** | 0.875 | 0.006 | -1.041 | 0.184 |
| **780 nm** | 0.874 | 0.011 | -0.924 | 0.225 |

Table 1. Deming regression parameters of PP vs. PaM calculations at different wavelengths.


| Wavelength | $C_{AE33,PP,day}(\lambda)$ | SE | $C_{AE33,PP,night}(\lambda)$ | SE | $C_{AE33,PaM,day}(\lambda)$ | SE | $C_{AE33,PaM,night}(\lambda)$ | SE |
|---|---|---|---|---|---|---|---|---|
| 470 nm | 3.56 | 0.06 | 3.49 (*) | 0.05 | 2.93 | 0.04 | 2.90 | 0.04 |
| 520 nm | 3.57 | 0.07 | 3.50 | 0.05 | 2.95 | 0.05 | 2.91 | 0.03 |
| 660 nm | 3.43 | 0.08 | 3.31 | 0.06 | 2.82 | 0.05 | 2.75 | 0.03 |
| 880 nm | 3.41 | 0.09 | 3.36 (**) | 0.07 | 2.79 | 0.06 | 2.77 | 0.04 |

Table 2: Multiple-scattering enhancement parameter and standard error (SE) for AE33 at different wavelength calculated separately on the day and night datasets using PP ($C_{AE33\_PP\_day}$ and $C_{AE33\_PP\_night}$, respectively) and PaM ($C_{AE33\_PaM\_day}$ and $C_{AE33\_PaM\_night}$, respectively) approaches. (*) original regression line intercept was 6.62 ± 2.15; (**) original regression line intercept was 4.48 ± 1.40.



| Wavelength | $C_{AE31,PP,day}(\lambda)$ | SE | $C_{AE31,PP,night}(\lambda)$ | SE | $C_{AE31,PaM,day}(\lambda)$ | SE | $C_{AE31,PaM,night}(\lambda)$ | SE |
|---|---|---|---|---|---|---|---|---|
| 470 nm | 4.42 | 0.10 | 4.25 | 0.08 | 3.65 | 0.07 | 3.53 | 0.06 |
| 520 nm | 4.38 | 0.10 | 4.18 | 0.08 | 3.61 | 0.06 | 3.48 | 0.05 |
| 660 nm | 4.44 | 0.11 | 4.18 | 0.08 | 3.65 | 0.07 | 3.48 | 0.05 |
| 880 nm | 4.34 | 0.13 | 4.12 | 0.09 | 3.55 | 0.08 | 3.39 | 0.06 |

Table 3: Multiple-scattering enhancement parameter and standard error (SE) for AE31 at different wavelength calculated separately on the day and night datasets using PP ($C_{AE31\_PP\_day}$ and $C_{AE31\_PP\_night}$, respectively) and PaM ($C_{AE31\_PaM\_day}$ and $C_{AE31\_PaM\_night}$, respectively) approaches.




| | Input data | 370 nm FF | 370 nm WB | 405 nm FF | 405 nm WB | 470 nm FF | 470 nm WB | 780 nm FF | 780 nm WB | 880 nm FF | 880 nm WB | 950 nm FF | 950 nm WB | $r_{WB}$ | $r_{FF}$ |
|---|---|---|---|---|---|---|---|---|---|---|---|---|---|---|---|
| *Calculation from $b_{abs}(\lambda)$ obtained using fixed multiple-scattering enhancement parameters: $C_{0,AE33}=1.57$, $C_{0,AE31}=2.14$* | AE33 7λ–fit | 49% | 51% | 51% | 49% | 54% | 46% | 66% | 34% | 68% | 32% | 70% | 30% | 0.94 | 0.88 |
| | AE31 7λ–fit | 48% | 52% | 51% | 49% | 54% | 46% | 65% | 35% | 68% | 32% | 70% | 30% | 0.93 | 0.88 |
| | AE33 370/950 | 56% | 44% | | | | | | | | | 76% | 24% | 0.94 | 0.92 |
| | AE31 370/950 | 51% | 49% | | | | | | | | | 72% | 28% | 0.94 | 0.91 |
| | AE33 370/880 | 50% | 50% | | | | | | | 70% | 30% | | | 0.94 | 0.91 |
| | AE31 370/880 | 52% | 48% | | | | | | | 71% | 29% | | | 0.94 | 0.91 |
| | AE33 470/950 | | | | | 68% | 32% | | | | | 80% | 20% | 0.94 | 0.91 |
| | AE31 470/950 | | | | | 60% | 40% | | | | | 75% | 25% | 0.95 | 0.91 |
| | AE33 470/880 | | | | | 61% | 39% | | | 74% | 26% | | | 0.95 | 0.91 |
| | AE31 470/880 | | | | | 63% | 37% | | | 76% | 24% | | | 0.94 | 0.91 |
| | AE33 4λ–fit $C_{0\_AE33}$ | | | | | 59% | 41% | 70% | 30% | 72% | 28% | | | 0.94 | 0.89 |
| | AE31 4λ–fit $C_{0\_AE31}$ | | | | | 65% | 35% | 75% | 25% | 77% | 23% | | | 0.93 | 0.91 |
| *Calculation from $b_{abs}(\lambda)$ obtained using optimised multiple-scattering enhancement parameters* | AE33 4λ–fit $C_{AE33,PP}$ | | | | | 68% | 32% | 78% | 22% | 80% | 20% | | | 0.93 | 0.90 |
| | AE31 4λ–fit $C_{AE31,PP}$ | | | | | 68% | 32% | 78% | 22% | 80% | 20% | | | 0.94 | 0.91 |
| | AE33 4λ–fit $C_{AE33,PaM}$ | | | | | 70% | 30% | 79% | 21% | 81% | 19% | | | 0.93 | 0.91 |
| | AE31 4λ–fit $C_{AE31,PaM}$ | | | | | 69% | 31% | 78% | 22% | 80% | 20% | | | 0.94 | 0.90 |
| | AE33 470/880 $C_{AE33,PP}$ | | | | | 68% | 32% | | | 79% | 21% | | | 0.92 | 0.90 |
| | AE31 470/880 $C_{AE31,PP}$ | | | | | 67% | 33% | | | 79% | 21% | | | 0.94 | 0.91 |
| | AE33 470/880 $C_{AE33,PaM}$ | | | | | 69% | 31% | | | 80% | 20% | | | 0.93 | 0.91 |
| | AE31 470/880 $C_{AE31,PaM}$ | | | | | 69% | 31% | | | 80% | 20% | | | 0.94 | 0.91 |
| | PP 405/780 | | | 65% | 35% | | | 77% | 23% | | | | | 0.83 | 0.89 |
| | PP 4λ–fit (*) | | | 62% | 38% | 65% | 35% | 75% | 25% | 76% | 24% | | | 0.81 | 0.87 |
| | PaM 405/780 | | | 68% | 32% | | | 80% | 20% | | | | | 0.82 | 0.90 |
| | PaM 4λ-fit (*) | | | 65% | 35% | 68% | 32% | 77% | 23% | 79% | 21% | | | 0.81 | 0.88 |

Header spanning: **Relative $b_{abs}(\lambda)$ source apportionment**


**Table 4. Absorption coefficient relative source apportionment using the Aethalometer model fixing $\alpha_{FF}=1$ and $\alpha_{WB}=2$. The model was applied to all available data using different data processing as presented in section 2.6. Values at 880nm for PP and PaM results were extrapolated. (*) 4-λ fit for PP and PaM data considers λ=405, 532, 635, 780 nm.**


| | Relative component apportionment (%) | | | | | | | | | |
|---|---|---|---|---|---|---|---|---|---|---|
| | 370 nm BC | 370 nm BrC | 470 nm BC | 470 nm BrC | 880 nm BC | 880 nm BrC | 950 nm BC | 950 nm BrC | $\alpha_{BrC}$ | $r_{BrC}$ |
| AE33 7λ–fit | 68% | 32% | 79% | 21% | 94% | 6% | 95% | 5% | 3.38±0.40 | 0.94 |
| AE31 7λ–fit | 65% | 35% | 75% | 25% | 91% | 9% | 92% | 8% | 2.99±0.56 | 0.91 |
| AE33 5λ–fit | | | 75% | 25% | 91% | 9% | | | 3.16±0.55 | 0.92 |
| AE31 5λ–fit | | | 80% | 20% | 95% | 5% | | | 3.66±0.97 | 0.94 |

**Table 5. Absorption coefficient relative component apportionment using the Aethalometer model fixing $\alpha_{BC}=1$. The model was applied to AE31 and AE33 data using different data processing as presented in section 2.7. The presented $r_{BrC}$ refers to 470 nm in all cases.**