# Peer review of "Determination of Aethalometer multiple-scattering enhancement parameters and impact on source apportionment during the winter 2017-2018 EMEP/ACTRIS/COLOSSAL campaign in Milan."

_Atmospheric Measurement Techniques, 2020_

## Referee Comment (RC1) · Anonymous Referee #1 · 8 Oct 2020

Review of the manuscript: "Determination of Aethalometer multiple-scattering enhancement parameters and impact on source apportionment during the winter 2017-2018 EMEP/ACTRIS/COLOSSAL campaign in Milan." By Bernardoni et al.

General comments: This manuscript presents an estimation of the filter multiple-scattering enhancement parameter at different wavelengths for AE31 and AE33 Aethalometers. The C parameter was obtained by comparing the ATN measurements

obtained with AE with off-line absorption measurements performed by means of the PP_UniMI polar photometer. Both PP and PaM approaches were tested. The effects of different instruments/$\alpha$exp on the Aethalometer source apportionment were also evaluated.

This work deals with an important aspect related to absorption measurement performed by means of Aethalometers which is the determination of the optimal C factor. The C is actually quite variable with time. The C is also site-dependent and the harmonization of the C factor looks quite complicated. This is why studies like the present one are potentially useful in order to contribute to our knowledge about the C factor. The strong advantage of the technique presented here is that the C was determined at 4 different wavelengths. And this is also important given that the C can be wavelength-dependent.

However, the C factors studied here are for the AE31 filter tape (quartz filter tape Pall Q250 quartz) and for the AE33 filter tape (TFE-coated glass fibre filter tape T60A20). My main concern is about the fact that the filter tapes characterized in this manuscript are quite out of date. The filter tape which should now be used in AE33 instruments is the tape M8060. The tape M8060 is, since 2017, the recommended filter tape. In this manuscript authors describe and characterize the multiple scattering artefacts for other tapes than the M8060.

The work cited in this manuscript (Drinovec et al., 2015) was published in 2015. But, recently, the tape M8060 (since 2017) has been indicated as the tape that should be used for ATN measurements with AE33. Many AE33 instruments in Europe actually use the tape M8060. Check for example Table 3 in this AMTD paper (https://amt.copernicus.org/preprints/amt-2020-344/amt-2020-344.pdf). Consequently, my main concern is about the relevance of this study given that the M8060 filter tape is not characterized here.

I suggest the authors to comment on this. The authors should explain why (and if) it is

important to characterize old filter tapes. Maybe the analysis presented here could be useful to harmonize previous measurements performed with the old tapes. The author should also mention that the filter M8060 is not studied here.

Despite this, the C values reported in this manuscript could be useful to correct previous AE measurements performed worldwide with AE31 and with AE33 deploying the old filter tape T60A20 (for AE33).

Specific comments: Page 2, Line 40-46: Mineral dust is also an important contributor to absorption, especially larger particles (>3-5 ïA■m) due to the high FeO2 content with consequent absorption in the UV and in the LW spectral ranges. Dust episodes are sporadic (but quite frequent actually in the Mediterranean Basin) and must be detected and removed from the database in order to properly characterize the UV BC or BrC absorption properties. Did you?

Pag 2, Line 45: "another possible light absorber". Please, remove "possible". Dust absorbs a lot of UV and LW radiation (especially larger particles). This is the reason why its effects on radiative forcing are not yet well constrained by global models.

Pag. 4, Line 105: It should be mentioned here that eBC at 880 nm is used for the quantification of BC concentrations.

Pag. 5, Line 138: "and correction does not affect data in terms of the absorption Ångström exponent". The authors should explain why the Sandradewi's algorithm does not affect the absorption Ångström exponent (AAE). If the algorithm is lambda-independent then, obviously, it does not affect the AAE. If it is lambda-independent, then, this is not necessarily a good thing because the correction could be ïĄň-dependent.

Pag.5, line 157: Here one of my major concern: The actual filter tape which should be used in AE33 is the tape M8060. The tape M8060 is the recommended filter tape. In this manuscript authors describe and characterize the multiple scattering artefact

for other tapes than the M8060. Is this true? The work cited in this manuscript (Drinovec et al.) was published in 2015. But, more recently, the tape M8060 has been indicated as the tape that should be used for measurements. Many AE33 in Europe actually use the tape M8060. Check Table 3 in the following AMTD paper (https://amt.copernicus.org/preprints/amt-2020-344/amt-2020-344.pdf). Consequently, my main concern is about the relevance of this study given that the M8060 filter tape is not characterized.

Paragraph 2.2.2, MAAP: MAAP data should be further corrected by applying a multiplicative factor of 1.05 to take into account that MAAP actually works at 637 nm and not at 670 nm. If the factor 1.05 is not taken onto account, then the MAAP data reported in this manuscript are at 670 nm (Müller et al., 2011).

Paragraph 2.6: Here different groups of wavelengths were used depending on the information available for each instrument. It would be useful for the reader if the authors could summarize these calculations in a table or in a block diagram.

Pag 13, Lines 397-408: Did the authors take into consideration that the AE software already includes $C_0$ values of 2.14 (or 1.57 depending on the AE used) in the calculations? If these $C_0$ values are not taken into account, then the experimental C will be overestimated.

Pag 13, Line 403: Why the intercept between PP and AE was not comparable to zero at 470 nm and 880 nm? Was it comparable when PaM was used? The authors suggest that a possible additive contribution from scattering could explain the non-zero intercepts. But if this is the reasons, why this effect was observed only at 470 and 880 nm?

Pag. 14, Line 402. The daytime C values were observed to be slightly higher compared to nighttime C. Is there any reason for this? Higher scattering (or SSA) during daytime?

Pag. 15, Line 445: Why the authors used C=2.14 (AE31) and C=1.57 (AE33) to obtain

the absorptions rather than using the experimental C values?

Paragraph 3.3: Is there any statistical significant difference among the C values obtained at the different wavelengths? Is the C parameter wavelength-dependent?

Paragraph 3.4; Figure 4: In relationship to Figure 4 the authors claim that "These graphs immediately show that different $\lambda$-dependence is present in data from different instruments.". I do not see a big difference considering that different wavelengths were used for PP and PaM compared to AE33 and AE31 in Figure 4. Moreover, the difference between AE33 and AE31 could be only due the compensation which is performed on-line by AE33. Note that almost no difference was reported between PP and PaM.

Pag. 15, Line 466: "The comparison of Fig. 5c and 5d to Fig. 4a as well as Fig. 5e and 5f to Fig. 4b, showed that the use of optimised multiplescattering enhancement parameters was not enough to harmonise the results of $\alpha$exp from different instruments.". I think that the main reason is that the optimized multiple-scattering parameters used are wavelength-independent thus these do not have a strong effect on the Absorption Angstrom exponent. Is this right?

---

## Referee Comment (RC2) · Anonymous Referee #2 · 1 Dec 2020

This manuscript addresses relevant scientific questions within the scope of AMT, notably allowing for a better evaluation of multiple-scattering enhancement parameters affecting multi-wavelength Aethalometer measurements. Its overall presentation (including the title, the abstract and the figures) is appropriate, clear and globally well structured. My only major concern is related to the timely relevance of obtained results, considering that the filter tape used here in relatively new AE33 device is no longer in use. This shall be stated much more clearly in the article. Authors should

also discuss the efficiency of this former type of filter tape for accurate BC source apportionment in light of the results they obtained here vs. reasons that eventually led the manufacturer to select another filter tape for AE33. In particular, do they consider that applying such a limited wavelength-dependence in C-AE33 values is fully sufficient to compensate for possible biases in BC source apportionment when using this type of filter tape ? And, conversely, what would be the quantitative impact of using a cosntant C-AE33 value in the present study ? As a minor comment, the mansucript would also benefit further explanations why the MWAA model could not be applied here to data obtained using the MWAA instrument, and, more importantly, if this is always the case ? Finally, authors could also clarify if they would generally recommend to use the PP_UniMI datasets with the PP approach or with the PaM approach.

---

## Author Comment (AC1) · 23 Dec 2020

*We thank the reviewer for comments, that were very helpful in improving the paper.*
*In the following, we answer point-by-point to the comments (blue lines).*

**Anonymous Referee #1**
Review of the manuscript: "Determination of Aethalometer multiple-scattering enhancement parameters and impact on source apportionment during the winter 2017- 2018 EMEP/ACTRIS/COLOSSAL campaign in Milan." By Bernardoni et al.
**General comments:** This manuscript presents an estimation of the filter multiple scattering enhancement parameter at different wavelengths for AE31 and AE33 Aethalometers. The C parameter was obtained by comparing the ATN measurements important to characterize old filter tapes. Maybe the analysis presented here could be useful to harmonize previous measurements performed with the old tapes. The author should also mention that the filter M8060 is not studied here. Despite this, the C values reported in this manuscript could be useful to correct previous AE measurements performed worldwide with AE31 and with AE33 deploying the old filter tape T60A20 (for AE33).
*Focusing on the Reviewer's main concern, in the updated version we tried to clarify the importance of the methodology –*
*that was here applied to the available dataset as case study, but can in principle be applied to any similar dataset,*
*independently of the tape used in AE33. Furthermore, we highlighted also that the results can by anyway of interest – even if*
*the specific kind of tape adopted in this work was discontinued – for sake of time-series harmonization (see corresponding*
*answer to detailed comments).*

**Specific comments:**
**Page 2, Line 40-46**: Mineral dust is also an important contributor to absorption, especially larger particles (>3-5 µm) due to the high FeO2 content with consequent absorption in the UV and in the LW spectral ranges. Dust episodes are sporadic (but quite frequent actually in the Mediterranean Basin) and must be detected and removed from the database in order to properly characterize the UV BC or BrC absorption properties. Did you?
*Looking at size distributions and back-trajectories, a Saharan dust event impacted the measurement site starting in the*
*evening on 27 January and ending in the early morning on 29 Jan.*
*An OPC operating at the site revealed the episode just in size classes with diameters > 2 µm (this size expected to be larger*
*when converted into aerodynamic diameter). The size-cut of the campaign was PM2.5, thus – if present - a small potential*
*impact of the event on our samples could be expected.*
*This was verified by comparing the results of regression lines with and without those samples. We report here the regression*
*line between AE31 and PP_UniMI for 470 nm (modified Fig. 3), as an example. We show in red the samples impacted by the*
*Saharan event. The samples of the Saharan dust event fall within all other samples and there is no difference in the slopes*
*with or without the events.*
*Results of the regression are reported to the third digit to highlight the absence of any statistically relevant difference.*

*Slope_all = 4.33±0.06*
*Slope_noSahara = 4.34±0.06*

[Figure]

**Pag 2, Line 45**: "another possible light absorber". Please, remove "possible". Dust absorbs a lot of UV and LW radiation (especially larger particles). This is the reason why its effects on radiative forcing are not yet well constrained by global models.
*Done.*

**Pag. 4, Line 105**: It should be mentioned here that eBC at 880 nm is used for the quantification of BC concentrations.
*Done.*

**Pag. 5, Line 138**: "and correction does not affect data in terms of the absorption Ångström exponent". The authors should explain why the Sandradewi's algorithm does not affect the absorption Ångström exponent (AAE). If the algorithm is lambda independent then, obviously, it does not affect the AAE. If it is lambda-independent, then, this is not necessarily a good thing because the correction could be lambda - dependent.
*The correction mentioned here is the Weingartner et al. (2003) correction for the loading effect which was implemented in a*
*special way in Sandradewi et al. (2008b). As no consensus in the scientific community is present on the need of wavelength-*

*dependent correction for Aethalometers (we hope to address this issue with the present manuscript), we wanted to point out that our approach preserved the original characteristics of the data.*

*Nevertheless, as the debate is still open, in the final version of the manuscript we modified as follows: (lines 136-138): "the loading term $R(ATN_{AE31})$ in Eq. (3) was dynamically determined following the Weingartner et al. (2003) algorithm as implemented in Sandradewi et al. (2008b) , and already used in previous heating-rate studies at the same site (Ferrero et al., 2018)".*

**Pag.5, line 157**: Here one of my major concern: The actual filter tape which should be used in AE33 is the tape M8060. The tape M8060 is the recommended filter tape. In this manuscript authors describe and characterize the multiple scattering artefact for other tapes than the M8060. Is this true? The work cited in this manuscript (Drinovec et al.) was published in 2015. But, more recently, the tape M8060 has been indicated as the tape that should be used for measurements. Many AE33 in Europe actually use the tape M8060. Check Table 3 in the following AMTD paper (https://amt.copernicus.org/preprints/amt-2020-344/amt-2020-344.pdf). Consequently, my main concern is about the relevance of this study given that the M8060 filter tape is not characterized.

*In the past few years, there has been much confusion about the different filter tapes for the AE33. During the campaign we have pragmatically used the filter type with published properties and known to perform well. We added the following sentence at lines 156-168: "In this work, the TFE-coated glass fibre filter tape T60A20 was used: it was the tape in use when AE33 was initially described (Drinovec et al., 2015). Due to discontinued production and supply of this filter tape, it should have been replaced by M8060. Nevertheless, there was considerable variation of the adoption of the last tape (M8060) by Aethalometer users, as seen from the instruments involved in the COST-COLOSSAL/ACTRIS inter-comparison campaign (Cuesta-Mosquera et al., 2020). To ensure accurate approach on the aerosol absorption measurements and reliable historical trend of such data, the filter tape characteristics need to be carefully investigated for all used filter tapes. While the filter tape used in the Aethalometer model AE31 is well characterised in the scientific literature, there is a lack of published research for the T60A20 filter tape. It is noteworthy that in a very recent paper on the analysis of data collected at the Global Atmosphere Watch (GAW) near-surface observatories, AE33 data were not analysed due to the lack of a unique value for converting the measured attenuation coefficient to particle light absorption coefficient (Laj et al., 2020). Thus, investigation on the T60A20 filter tape will ensure continuity towards a better harmonization in the timeseries of measurements by AE33. Furthermore, the methodology presented in this paper which can be similarly applied to any other dataset and thus can give an important contribution to the currently open scientific debate on the determination of aerosol absorption properties."*

**Paragraph 2.2.2, MAAP:** MAAP data should be further corrected by applying a multiplicative factor of 1.05 to take into account that MAAP actually works at 637 nm and not at 670 nm. If the factor 1.05 is not taken onto account, then the MAAP data reported in this manuscript are at 670 nm (Müller et al., 2011).

*We are aware of the paper by Müller et al., 2011 and of the data treatment implemented in ACTRIS.*
*Nevertheless:*
*- only one MAAP was used in Sheridan et al., 2005 for comparison to reference ext-minus-sca and photoacoustic*
*- in different comparisons (including the ACTRIS inter-comparison workshop in 2013), relative sensitivities among MAAPs after flow calibration – compared to a reference - was in the range -3.8% to 5.6% - within the range of "disagreement" claimed in Müller et al., 2011 referring to Sheridan et al., 2005.*
*It is noteworthy that even multiplying MAAP values by 1.05, agreement between MAAP and PP_UniMI using PaM approach is within few percent.*

[Figure]

**Paragraph 2.6:** Here different groups of wavelengths were used depending on the information available for each instrument. It would be useful for the reader if the authors could summarize these calculations in a table or in a block diagram.

*We added Table 1 to summarise all the tests described in par. 2.6 and we added the following sentence (lines 320-322): "A summary of all the performed tests, for each instrument and b_abs measurement methodology, in terms of input*

*wavelengths and of the wavelengths of analysed output data for both 2-λ and multi-λ fit Aethalometer model approaches can be found in Table 1".*

**Pag 13, Lines 397-408:** Did the authors take into consideration that the AE software already includes C_0 values of 2.14 (or 1.57 depending on the AE used) in the calculations? If these C_0 values are not taken into account, then the experimental C will be overestimated.

*$b_{ATN}(\lambda)$ for AE33 was derived from $b_{abs}$ data considering $C_0=1.57$.*
*As for AE31, $C_0=2.14$ was not considered as it is already included in the $\sigma_{AE31}$ used in the algorithm (see equation 3 for $b_{ATN\_AE31}$ retrieval) (it would be otherwise double-counted). Indeed, it should be noticed that the mass attenuation cross-section $\sigma_{AE31}(\lambda)$ used in the AE31 algorithm and the mass absorption cross-section $\sigma_{abs}(\lambda)$ (of BC as used in the AE33) algorithm differ exactly for a 2.14 value.*

**Pag 13, Line 403:** Why the intercept between PP and AE was not comparable to zero at 470 nm and 880 nm? Was it comparable when PaM was used? The authors suggest that a possible additive contribution from scattering could explain the non-zero intercepts. But if this is the reasons, why this effect was observed only at 470 and 880 nm?

*As now reported at lines 412-418: "The intercepts at 470 nm and 880nm can be related to different effects (or combination of them). It has to be considered that few Mm-1 represent the limit of detection for PP_UniMI, thus it may have a role on the intercept. Furthermore, Valentini et al., (2020) performed sensitivity tests about the role of asymmetry parameter on results by PP and PaM approaches. These tests showed few percent variation in the results moving from g = 0.50 to g = 0.75 and intercepts about 0.010±0.001 in units of absorbance (1–ω)·τ. Finally, the approach presented in Eq. (1) neglects a possible additive contribution from scattering (i.e. is best at low single scattering albedo - SSA). Bias of such an approximation – possibly depending on the wavelength – can contribute to the observed intercepts".*

**Pag. 14, Line 402.** The daytime C values were observed to be slightly higher compared to nighttime C. Is there any reason for this? Higher scattering (or SSA) during daytime?

*Higher scattering could contribute to a higher scattering artefact in the transmission measurements and lead to higher C values. Unfortunately, parallel measurements of scattering properties were not available, so we do not have experimental data supporting this hypothesis. It could also be related to different shape-size of daytime particle compared to night-time one (e.g. resuspension is generally higher during daytime. Big particles enhance forward scattering, maybe increasing the fraction of light impinging on the filter allowing enhanced multiple scattering effects). Nevertheless, this effect should be limited by the size-cut used in this campaign (PM2.5). These considerations were added to the text (lines 441-446): "Possible reasons for higher daytime values compared to night-time ones could be differences in particle SSA. Also, different size-distribution can play a role. As an example, a higher fraction of bigger particles – e.g. related to resuspension – can enhance forward scattering, thus increasing the fraction of light impinging on the filter. Nevertheless, the first hypothesis would have required parallel scattering measurements to be supported and the second should give limited effect related to the size-cut (PM2.5) used in this campaign. Anyway, further experimental information should be collected in future similar campaigns to clarify this aspect."*

**Pag. 15, Line 445:** Why the authors used C=2.14 (AE31) and C=1.57 (AE33) to obtain the absorptions rather than using the experimental C values?

*This was made to show the AAE frequency distribution for not-corrected Aethalometer data. Distributions obtained using experimental C values are subsequently shown in Figure 5. The following sentence was added: "This figure should be considered as reference for the results obtained by a routine analysis."*

**Paragraph 3.3:** Is there any statistical significant difference among the C values obtained at the different wavelengths? Is the C parameter wavelength-dependent?

*From the results, only weak wavelength-dependence can be claimed for AE33 and no wavelength-dependence for AE31. This is now better stated in the conclusions (lines 590-594): "Comparing $C_{AE31}(\lambda)$ at different wavelengths with all approaches for daytime and night-time data, they were all within standard error (SE) for AE31 and no statistically significant wavelength-dependence was found in our work. For AE33, results at 470 and 520 nm are not comparable to those obtained at 660 and 880 nm within SE: this suggests a weak wavelength-dependence. Nevertheless, if 3·SE is considered as limit for statistically significant differences, then also for AE33 no statistically significant wavelength-dependence can be claimed". Furthermore, a brief sentence (missing in the previous version) was added at lines 425-428 for AE33: "It is noteworthy that values at 470 nm and 520 nm were comparable within SE and the same occurs for the values at 660 nm and 880 nm for both PP and PaM approaches pointing to a weak wavelength dependence. Nevertheless, if 3·SE is considered for statistically significant differences, all the values were comparable and no wavelength-dependence can be claimed."*

**Paragraph 3.4; Figure 4:** In relationship to Figure 4 the authors claim that "These graphs immediately show that different _-dependence is present in data from different instruments.". I do not see a big difference considering that different wavelengths were used for PP and PaM compared to AE33 and AE31 in Figure 4. Moreover, the difference between AE33

and AE31 could be only due the compensation which is performed on-line by AE33. Note that almost no difference was reported between PP and PaM.
*The sentence was removed as the description of the figure.*

**Pag. 15, Line 466:** "The comparison of Fig. 5c and 5d to Fig. 4a as well as Fig. 5e and 5f to Fig. 4b, showed that the use of optimised multiple scattering enhancement parameters was not enough to harmonise the results of _exp from different instruments.". I think that the main reason is that the optimized multiple-scattering parameters used are wavelength-independent thus these do not have a strong effect on the Absorption Angstrom exponent. Is this right?

*We considered C($\lambda$) in the correction, even if it was not statistically significant. Thus, even if the contribution was expected to be limited, we tried to verify whether it was sufficient for instrument harmonization.*

*This was not the case and the possible reasons are mentioned at lines 482-489: "First of all, the measured absorption coefficients are the sum of (at least) two contributions (traffic, biomass burning) featuring different absorption Ångström exponents, thus the analytical dependence of their sum is not expected to be exactly exponential. Second, cross-sensitivity to scattering is expected to be an additive term, which is neglected in the approach presented in Eq. (1), which approximates the relationship between absorption and extinction by the use of a single multiplicative factor. Third, we are considering average factors and applying them to all the dataset, whereas sample-by-sample differences are expected, e.g. in the scattering properties of the particles. Finally, it should be recalled that PP_UniMI wavelengths were 405, 532, 635 and 780 nm, whereas the wavelengths considered for Aethalometers 4-$\lambda$ calculations were 470, 520, 660, and 880 nm".*

---

## Author Comment (AC2)

*We thank the reviewer for comments, that were very helpful in improving the paper.*
*In the following, we answer point-by-point to the comments (blue lines).*

**Anonymous Referee #2**
This manuscript addresses relevant scientific questions within the scope of AMT, notably allowing for a better evaluation of multiple-scattering enhancement parameters affecting multi-wavelength Aethalometer measurements. Its overall presentation (including the title, the abstract and the figures) is appropriate, clear and globally well structured. My only major concern is related to the timely relevance of obtained results, considering that the filter tape used here in relatively new AE33 device is no longer in use. This shall be stated much more clearly in the article.

*Focusing on the Reviewer's major concern, in the updated version we tried to clarify the importance of the methodology – that was here applied to the available dataset as case study, but can in principle be applied to any similar dataset, independently of the tape used in AE33. Furthermore, we highlighted also that the results can by anyway of interest – even if the specific kind of tape adopted in this work was discontinued – for sake of time-series harmonization.*
*As now mentioned at lines 156-168:*
*"In this work, the TFE-coated glass fibre filter tape T60A20 was used: it was the tape in use when AE33 was initially described (Drinovec et al., 2015). Due to discontinued production and supply of this filter tape, it should have been replaced by M8060. Nevertheless, there was considerable variation of the adoption of the last tape (M8060) by Aethalometer users, as seen from the instruments involved in the COST-COLOSSAL/ACTRIS inter-comparison campaign (Cuesta-Mosquera et al., 2020). To ensure accurate approach on the aerosol absorption measurements and reliable historical trend of such data, the filter tape characteristics need to be carefully investigated for all used filter tapes. While the filter tape used in the Aethalometer model AE31 is well characterised in the scientific literature, there is a lack of published research for the T60A20 filter tape. It is noteworthy that in a very recent paper on the analysis of data collected at the Global Atmosphere Watch (GAW) near-surface observatories, AE33 data were not analysed due to the lack of a unique value for converting the measured attenuation coefficient to particle light absorption coefficient (Laj et al., 2020). Thus, investigation on the T60A20 filter tape will ensure continuity towards a better harmonisation in the timeseries of measurements by AE33. Furthermore, the methodology presented in this paper which can be similarly applied to any other dataset and thus can give an important contribution to the currently open scientific debate on the determination of aerosol absorption properties"*

Authors should also discuss the efficiency of this former type of filter tape for accurate BC source apportionment in light of the results they obtained here vs. reasons that eventually led the manufacturer to select another filter tape for AE33. In particular, do they consider that applying such a limited wavelength-dependence in C-AE33 values is fully sufficient to compensate for possible biases in BC source apportionment when using this type of filter tape ? And, conversely, what would be the quantitative impact of using a constant C-AE33 value in the present study?

*We would like to point out that the tape used in this work was the T60A20, which was simply discontinued.*
*We did not investigate M8050 tape which was chosen after T60A20, but was abandoned and replaced by the M8060 because of issues concerning filter loading correction at lower wavelengths.*
*Considerations on the role of optimized $C_{AE33}$ on source apportionment harmonization was mentioned in par 3.5 (see lines 527-529 in the previous version, now lines 542-544), but not in the conclusions. So now we added the following sentence at Lines 609-611: "However, relative apportionment agreed within 5% at most (and, more in detail, PP_UniMI source apportionments results were always within the variability of Aethalometers results by different approaches) when optimised multiple-scattering enhancement parameters were considered for Aethalometers."*

As a minor comment, the manuscript would also benefit further explanations why the MWAA model could not be applied here to data obtained using the MWAA instrument, and, more importantly, if this is always the case ?

*MWAA model can be applied whenever information at least at 5-λ is available (Bernardoni et al., 2017b). Thus, it was not applicable to our dataset because photometric off-line measurements were carried out by PP_UniMI that was a 4-λ instruments at the time of the measurements. Opposite, the MWAA instrument (Massabò et al., 2015) is a 5-λ instrument. Thus, the MWAA model can be applied to data collected using the MWAA instrument, indeed.*
*This is now better clarified at lines 341-344: " Nevertheless, tests showed issues with numerical calculation when using only 4-λ information (i.e. lack of convergence and/or fit parameter instability) and a minimum of 5-λs is necessary to ensure model stability (Bernardoni et al., 2017b). Thus, in this work the MWAA model was run only using Aethalometer data as input (PP_UniMI is a 4-λ instrument)".*

Finally, authors could also clarify if they would generally recommend to use the PP_UniMI datasets with the PP approach or with the PaM approach.

*We added the following sentence in the conclusion, to better highlight why we presented both results (lines 576-579)*
*"We provided both results as the MAAP is often used as a reference instrument, and multiple-scattering enhancement*

*parameters can be directly compared to others present in the literature. Nevertheless, PP_UniMI performs a more detailed analysis by measuring the phase function in the in the scattering plane, in principle improving the accuracy of the measurements."*